



# New insight from CryoSat-2 sea ice thickness for sea ice modelling

David Schröder[1], Danny L. Feltham[1], Michel Tsamados[2], Andy Ridout[2] and Rachel Tilling[3]

[1]Centre for Polar Observation and Modelling, Department of Meteorology, University of Reading, Reading, RG6 6BB, U.K.
[2]Centre for Polar Observation and Modelling, Department of Earth Sciences, University College London, London, WC1E 6BT, U.K.
[3]Centre for Polar Observation and Modelling, School of Earth and Environment, University of Leeds, Leeds, LS2 9JT, U.K.

*Correspondence to*: David Schröder (D.Schroeder@reading.ac.uk)

**Abstract.** Estimates of Arctic sea ice thickness are available from the CryoSat-2 (CS2) radar altimetry mission during ice growth seasons since 2010. We derive the sub-grid scale ice thickness distribution (ITD) with respect to 5 ice thickness categories used in a sea ice component (CICE) of climate simulations. This allows us to initialize the ITD in stand-alone simulations with CICE and to verify the simulated cycle of ice thickness. We find that a default CICE simulation strongly underestimates ice thickness, despite reproducing the inter-annual variability of summer sea ice extent. We can identify the underestimation of winter ice growth as being responsible and show that increasing the ice conductive flux for lower temperatures (bubbly brine scheme) and accounting for the loss of drifting snow results in the simulated sea ice growth being more realistic. Sensitivity studies provide insight into the impact of initial and atmospheric conditions and, thus, on the role of positive and negative feedback processes. During summer, atmospheric conditions are responsible for 50% of September sea ice thickness variability through the positive sea ice and melt pond albedo feedback. However, atmospheric winter conditions have little impact on winter ice growth due to the dominating negative conductive feedback process: the thinner the ice and snow in autumn, the stronger the ice growth in winter. We conclude that the fate of Arctic summer sea ice is largely controlled by atmospheric conditions during the melting season rather than by winter temperature. Our optimal model configuration does not only improve the simulated sea ice thickness, but also summer sea ice concentration, melt pond fraction, and length of the melt season. It is the first time CS2 sea ice thickness data have been applied successfully to improve sea ice model physics.

## 1 Introduction

Historical observations of sea ice thickness have been limited due to their sparse spatial and temporal coverage of, and uncertainties in, measurements. Prior to the launch of the European Space Agency's (ESA) first European Remote Sensing satellite (ERS-1) in 1991, most data were collected from submarines operating beneath the Arctic pack ice. Upward-looking sonars measure the submerged portion of the ice (draft), which can be converted to thickness by making assumptions about the snow depth and the densities of ice, snow and water. Based on sea ice draft observations from 34 US Navy submarines, a decrease of mean autumn sea ice thickness from 2.8 m to 1.6 m could be identified over the period 1975 to 2000 within the central Arctic Ocean (Rothrock et al., 2008). While the accuracy and the spatial coverage was sufficient to give evidence of sea ice thinning in the Arctic and to provide a basis for simulating the trend, these data are of limited use for evaluating the spatial and temporal variability of sea ice across the Arctic, and in climate models. More recently, cryosphere-focused satellite altimeters such as the NASA Ice, Cloud, and land Elevation Satellite (ICESat) and ESA CryoSat-2 (CS2) have allowed estimation of sea ice thickness across the Arctic (Giles et al., 2007 & 2008, Kwok and Rothrock, 2009, Laxon et al., 2013).

The performance of sea ice models in climate models is most commonly evaluated by using Arctic- and Antarctic-wide sea ice extent and sea ice area data from passive microwave data and Arctic sea ice volume from the Pan-Arctic Ice Ocean Modeling and Assimilation System PIOMAS (Zhang and Rothrock, 2003; e.g. Massonnet et al., 2012; Stroeve et al., 2012; Rae et al., 2014; Shu et al., 2015; Ridley et al., 2018). PIOMAS is a reanalysis product, which does not include ice thickness observations. Stroeve et al. (2014) combined submarine, aircraft and satellite data to evaluate sea ice thickness in climate model simulations. Their assessment revealed shortcomings regarding spatial patterns of sea ice thickness that were larger



than uncertainties in the verification data. CS2 sea ice thickness data have recently been used to initialize sea-ice ocean and climate models to improve seasonal predictions (Allard et al., 2018; Blockley et al., 2018).

In this study, we aim to improve sea ice physics and to calibrate poorly constrained parameters in the sea ice model CICE using the CS2 ice thickness data. Given CS data are available from October to April, our focus lies on processes controlling

winter ice growth. We note that several factors contribute to CS2-derived sea ice thickness uncertainties, including the assumption that the radar return is from the snow/ice interface (*Willat et al.*, 2011), snow depth departures from climatology, and the use of fixed snow and ice densities. To enable a meaningful comparison of sea ice thickness between CICE and CS2, we consider the strengths and weaknesses of the model and the data. We design sensitivity experiments to close the gap between our default CICE simulation and CS2 as well as to investigate the impact of initial conditions and atmospheric forcing

data. The latter experiments address questions regarding the impact of the last three exceptionally warm Arctic winters on the sea ice decline (Stroeve et al., 2018) and gives new insight into the strengths of positive and negative feedback mechanisms that govern the evolution of sea ice.

## 2 Ice Thickness Distribution from CryoSat-2

The CS2 radar altimetry mission was launched in April 2010, providing estimates of ice thickness during the ice growth season

from October to April. During summer the formation of melt ponds interferes with the radar signal retarding accurate measurements. As for the derivation of sea ice thickness from ice draft using submarine data, the freeboard (the height of sea ice above the water surface) is estimated from the satellite data and can be converted to thickness by assuming hydrostatic equilibrium and applying values for the densities of ice, snow and water as well as the snow depth. The principal challenges in deriving an accurate sea ice thickness using satellite altimetry are the discrimination of ice and open water, retracking radar

waveforms to obtain height estimates, constructing sea surface height beneath the ice, and estimating the depth of the snow cover. Ice thickness is retrieved from freeboard by processing CS2 Level 1B data, with a footprint of approximately 300 m by 1700 m (Wingham et al., 2006), and assuming snow density and snow depth from the *Warren et al*. (1999) climatology (hereafter *W99*), modified for the distribution of multi-year versus first-year ice (see *Laxon et al*., 2013 and *Tilling et al.*, 2018 for data processing details).

In this study, we bin the individual thickness point measurements provided by the Centre for Polar Observation and Modelling (CPOM) into 5 CICE thickness categories, (1) ice thickness $h < 0.6$ m, (2) $0.6$ m $< h < 1.4$ m, (3) $1.4$ m $< h < 2.4$ m, (4) $2.4$ m $< h < 3.6$ m, (5) $h > 3.6$ m, on a rectangular 50 km grid for each month. The mean area fraction and mean thickness is derived for each thickness category and these values are interpolated on the ORCA tripolar 1deg grid used by CICE (~40km grid resolution). Grid points with less than 100 individual measurements and a mean sea ice thickness of less than 0.5 m are omitted

due to their increased uncertainty (Ricker et al., 2017). Otherwise, all individual observations from applied grid points are included. Negative thickness values are retained in the CS2 processing to prevent statistical positive bias of the thinner ice and are added to category 1. This approach allows us to initialize the CICE model with the full ice thickness distribution (ITD) rather than to derive the ITD from the mean sea ice thickness (Hunke et al., 2015). A realistic ITD is critical for simulating ice growth and ice melt rates correctly: given the identical mean ice thickness, a wider distribution leads to an increase of ice

growth in winter, because ice growth mainly takes place over the thinner ice; whereas during summer, the thinner ice melting away increases the lead fraction, resulting in a warmer ocean mixed-layer temperature, thus accelerating the sea ice melt.

In addition to CPOM CS2, we include sea ice thickness data provided by the Alfred Wegener Institute AWI (*Hendricks et al.*, 2016) and the National Aeronautics and Space Administration NASA (*Kurtz and Harbeck*, 2017). The comparison of the three CS2 data sets illustrates a part of their uncertainty and helps us to assess the robustness of model-CS2 ice thickness differences.

While all three data providers rely on *W99* for snow depth and density, there is variation in how it is applied. CPOM spatially average the *W99* snow depth and then halve it over first-year ice (Tilling et al., 2018), AWI also halve *W99* snow depth over



first-year ice but discard measurements in lower latitude regions (Ricker et al., 2014; Hendricks et al., 2016), and NASA produce a blended snow depth dataset by also including satellite, reanalysis, and airborne estimates (Kurtz et al., 2013). Each institution also processes the radar returns differently. When estimating ice freeboard the range to the main scattering horizon of the radar return is obtained using a retracker algorithm. CPOM use a Gaussian-exponential retracker for ocean waveforms

and a threshold first maximum retracker algorithm (TFMRA) with 70% threshold for sea ice waveforms (Tilling et al., 2018), AWI apply a 50% TFMRA to all waveforms (Ricker et al., 2014; Hendricks et al., 2016), and the NASA estimates rely on a physical model to best fit each waveform (Kurtz et al., 2014; Kurtz and Harbeck, 2017). This could lead to ice thickness differences based on different retrackers and thresholds applied.

CS2 ice thickness data for October appear to be less robust as indicated by large differences between the three products for

thin first-year ice (not shown) and by large differences in comparison with ESA's Soil Moisture and Ocean Salinity (SMOS) sea ice data (Wang et al., 2016), so we only include CS data from November to April in our study.

## 3 Model setup und simulations

### 3.1 Setup

CICE is a dynamic-thermodynamic sea-ice model designed for inclusion within a global climate model. Here, we perform

stand-alone (fully forced) CICE simulations with version 5.1.2 for a pan-Arctic region (~40 km grid resolution). CICE contains a simple mixed-layer ocean model with a prognostic ocean temperature. To account for heat transport in the ocean, we restore the mixed-layer ocean temperature and salinity to climatological monthly means from MYO-WP4-PUM-GLOBAL-REANALYSIS-PHYS-001-004 (Ferry et al., 2011) with a restoring timescale of 20 days. We apply a climatological ocean current (monthly means) from the same ocean reanalysis product. NCEP Reanalysis-2 (NCEP2) atmospheric reanalysis data

(*Kanamitsu et al.*, 2002, updated 2017) are used as atmospheric forcing. We perform a multi-year simulation from 1980 to 2017 which does not utilise CS2 ice thickness data (referred to as CICE-free) and seven, 1-year long simulations which are initialized with CS2 ice thickness data (referred to as CICE-ini) starting in mid-November and running until the end of November of the following year for the 7 winter periods from 2010/2011 to 2016/2017. The initial thickness and concentration for each of the 5 ice categories is taken from the CPOM CS2 ITD November mean thickness fields. For grid points without

CS2 data, and for all other variables (e.g. temperature profiles, snow volume), results from the free CICE simulation are applied. In this way, CICE simulations cover the pan-Arctic region, but in regions where no CS2 are available, we restart SIT values from the free CICE model run. While this approach would be problematic in a coupled model, in a stand-alone sea ice simulation the model adjustment to the new conditions is smooth and the impact of using the vertical temperature profile from the free simulation only affects sea ice thickness on the order of millimeters.

### 3.2 Reference simulation (CICE-default)

Our reference simulation includes a prognostic melt pond model (Flocco et al., 2010 & 2012) and the elastic anisotropic plastic rheology (Wilchinski and Feltham, 2006; Tsamados et al., 2013; Heorton et al., 2018). Otherwise, default CICE settings are chosen: 7 vertical ice layers, 1 snow layer, linear remapping ITD approximation (Libscomp and Hunke, 2004), Bitz and Libscomp (1999) thermodynamics, Maykut and Untersteiner (1971) conductivity, Rothrock (1975) ridging scheme with a Cf

value of 12 (empirical parameter that accounts for frictional energy dissipation) and the Delta-Eddington radiation scheme (Briegleb and Light, 2007).



### 3.3 Sensitivity simulations

### 3.3.1 Improving sea ice thickness

Comparing the CICE simulation with CS2 reveals that CICE default underestimates the mean monthly sea ice thickness by about 0.8 m (see Fig. 1a and discussion in Section 4). This motivates sensitivity experiments with the aim to increase ice

thickness. All sensitivity simulations are listed and described in Table 1. They have been performed as multi-year simulations (1980 to 2017, CICE-free) and seven 1-year-long simulations initialized with CS2 ice thickness (CICE-ini).

### 3.3.2 Uncertainty in atmospheric forcing data

We perform two sensitivity experiments to explore the impact of uncertainty in atmospheric forcing on simulated sea ice conditions: decrease of incoming longwave radiation by 15% and decrease of 2m-air temperature by 2 K. For both experiments

seven 1-year-long simulations are performed which are initialized with CS2 ice thickness (CICE-ini) in mid-November. The setup of the reference run has been applied (CICE-default). See Table 2 for details.

### 3.3.3 Impact of initial conditions and atmospheric forcing data

We perform additional sensitivity experiments to explore the impact of initial and forcing conditions. Therefore, we conduct simulations with climatological initial and forcing conditions. For each experiment seven, 1-year-long simulations are

performed that are initialized with CS2 ice thickness (CICE-ini) in mid-November. The setup of the most realistic configuration has been applied (CICE-best). See Table 3 for details and Section 4.4 for deriving our most realistic configuration.

## 4 Results

### 4.1. Defining region for comparing model sea ice thickness with CryoSat-2 data

Several factors lead to errors in ice thickness retrieval from CS2, in particular the assumption of a climatological snow depth. The resulting ice thickness error is about 5 times larger than the error in snow depth, e.g. an underestimation in snow depth of 0.1 m leads to an underestimation in ice thickness of 0.5 m for a typical ice freeboard of 0.2 m (Tilling et al., 2015).

To enable a meaningful comparison of simulated sea ice thickness between CICE and CS2, we have to reduce the impact of errors. While for individual years and regions the $W99$ snow load can differ from reality by more than 0.1 m (Warren et al.,

1999), the average snow conditions are accurately represented over multi-year ice (e.g. Haas et al., 2017). Therefore, we apply a climatology for the CS2 period 2010 to 2017. Further, we compare spatial averages of ice thickness to reduce the impact of random errors.

We select a region over which to compare model simulations with CS2 data, for which CS2 data exist for all winter periods with at least 100 single observations per month and grid cell, and a minimum mean ice thickness of 0.5 m. Focusing on winter

ice growth, we limit the region even further to grid cells in which ice growth is dominating over the impact of dynamics on sea ice thickness change. The mean simulated ice growth from November to April (2010 to 2016, see Fig. 2a) varies between 0.6 and 1.0 m in the Central Arctic and can reach values of more than 2 m in the polynya regions along the Siberian coast (Dmitrenko et al., 2009; Bauer et al., 2013). Ice thickness is also modified by dynamical processes such as advection, convergence and ridging. While the total impact is below +/- 0.25 m in the Central Arctic, more than 2 m of ice is exported on

average from some coastal regions and transported to the Fram Strait (see Fig. 2b). For our region of interest, we only select grid cells in which the impact of dynamics on sea ice thickness change during winter is less than 0.25 m in magnitude. The resulting comparison region restricted by CS2 observations and dynamical sea ice change is shown in Fig. 3. We use this region for the rest of the paper.





### 4.2 Comparison of default CICE simulation with CryoSat-2 data

The mean annual cycle of ice thickness over our region of interest is compared in Fig. 1 with CS2. CICE-free-default underestimates the mean monthly sea ice thickness by about 0.8 m. It is noteworthy that this simulation is generally realistic in comparison to SSM/I sea ice extent (see Fig. 4) and represents the inter-annual variability of e.g. the September ice extent.

The magnitude of the ice thickness difference of 0.8 m cannot be explained by the uncertainty of the CS2 ice thickness alone (Tilling et al, 2016; Stroeve et al., 2018) and so indicates a model error.

CS2 ITD can be applied to initialize a CICE simulation with the observed ice thickness and CS2 data enable us to trace the ice thickness continuously through the whole winter until April. We apply the mean November CS2 ITD to initialize the CICE simulations in mid-November for the years from 2010 to 2016. Starting in mid-November, the mean simulated April ice

thickness from CICE-ini-default is about 0.25 m too thin (Fig. 1b). This indicates that the winter ice growth is underestimated in the model. To explore the reasons for the underestimation, we will first examine the impact of atmospheric forcing data (Section 4.3) and then the impact of the physical processes involved and how they are represented in CICE (Section 4.4).

### 4.3 Impact of uncertainty in atmospheric forcing data

CICE ice growth in the central Arctic mainly depends on atmospheric forcing (in particular incoming longwave radiation and

air temperature), the parameterization of the turbulent atmospheric heat fluxes (heat transfer coefficients) and the conductive heat fluxes within the ice and snow layers. While the impact of the turbulent ocean heat flux under the ice can be large in the marginal ice zone, the ocean temperature is generally close to the freezing temperature in the central Arctic during winter.

To explore to what extent the underestimation of ice thickness can be attributed to errors in atmospheric forcing data, we performed two sensitivity studies in which we decreased (1) the incoming longwave radiation by 15 W m$^{-2}$ (CICE-Ldown15-

ini) and (2) the 2m-air-temperature (CICE-Tair2-ini) during the whole simulation period and for every location (see Table 2). We have chosen these values as estimates for potential total systematic atmospheric errors (e.g. Chaudhuri et al., 2014). The impact of these atmospheric perturbations on ice growth is small (see. Fig. 5) and the gap between CS2 ice thickness and CICE-default ice thickness can only be reduced marginally. The reduced longwave radiation and air temperature lead to a reduction of ice surface temperature in the range of 2 to 3 K and only increase ice growth by about 5% each. This is due to the

fact that surface temperatures are around -30 °C during winter and the conductivity of the snow layer is low. Winter ice growth is not strongly affected by errors in atmospheric conditions. This is fundamentally different during the melting season. Fig. 5 reveals that the sea ice would be 0.9 m thicker in September due to the reduction of incoming longwave radiation and 1.4 m thicker due to the decrease of 2m-air-temperature starting with the same initial conditions in the November of the previous year. These sensitivity studies reveal that the underestimation of winter ice growth cannot be explained by errors in atmospheric

forcing data.

### 4.4 Improving CICE simulation by varying model physics

In the sea ice model, local ice thickness changes are calculated by thermodynamic processes (ice growth and melt) and dynamic processes (advection and ridging). The thermodynamic change depends on the energy balance at the interfaces between atmosphere, snow, ice and ocean derived from the shortwave and longwave radiation fluxes, the turbulent heat fluxes and the

conductive heat flux through the ice and snow. Addressing the individual contributions systematically, we alter model physics within their range of uncertainty with the aim to increase sea ice thickness. Our model changes are presented in a cumulative way.

The shortwave radiation budget strongly depends on the surface albedo parameterization. Here, we indirectly increase the total ice surface albedo (accounting for snow and pond covered as well as bare ice) by releasing more melt water into the ocean,

and hence reducing the formation of ponded (darker) water over the surface of the ice (CICE-mw, see Table 1). While there are several possibilities to increase albedo values, we selected the release of melt water because of its impact on the albedo





feedback mechanism and the poor knowledge about realistic assumptions. The impact on the simulated melt pond fraction can be seen in Fig. 6 for July 2012. Comparing our simulated melt pond fraction with satellite products shows that the mean July pond fraction of CICE-mw (25 %) is closer to the mean values based on MERIS (Istomina et al., 2015) and MODIS data (Roesel et al., 2012) (24 % for both) than the mean pond fraction of CICE-default (28 %). Furthermore, the RMS error with

respect to MERIS is reduced from 16 % (CICE-default) to 14 % (CICE-mw) justifying our increased release of melt water. Fig. 1a shows the thickness error is marginally reduced by CICE-mw: from about 0.8 m to 0.7-0.75 m. As expected there is no impact during winter for the initialized simulation, but the summer ice is 0.2 m thicker (Fig. 1b).

In the next sensitivity experiment, we address the sea ice advection and the turbulent heat fluxes. We apply the form drag parameterization of Tsamados et al., (2014) in addition to the release of more melt water (CICE-mw-form). Ice thickness

increased slightly with respect to CICE-mw due to a reduced ice drift speed resulting in a weaker ocean-ice heat flux and less ice export. In addition, increasing the emissivity of snow and ice from 0.95 to 0.976 (CICE-mw-form-e) affects the longwave radiation budget. The increase reduces summer melt by a few centimeters, but no impact during winter is visible.

Due its low conductivity snow depth controls the conductive heat flux. Here, we implement a snow drift scheme based on Lecomte et al. (2014) which reduces the snow depth by 20 to 40% (CICE-mw-form-e-sd, see Fig. 7) and has the biggest impact

on sea ice thickness from all individual changes. With respect to CICE-default, the thickness error has been reduced from 0.8 m to 0.25 m (see CICE-mw-form-e-sd-free in Fig. 1a). The reduction of snow depth leads to a strong increase in ice growth in winter, but also to a moderate increase of summer melt due to an earlier disappearance of snow. This can be seen comparing CICE-mw-form-e-sd-ini with CICE-mw-form-e-ini in Fig. 1b. The reduction of snow leads to an increase of May ice thickness of 0.12 m and to a decrease of September ice thickness of 0.06 m. Applying an increased conductivity coefficient for colder

temperature (Pringle et al., 2007, CICE-mw-form-e-sd-bubbly) in addition reduces the error for CICE-free and CICE-ini to less than 0.1 m. This modification increases winter ice growth, but it has no impact during summer.

So far, we compared the mean annual cycle of sea ice thickness over the period from 2010 to 2017. Fig. 8 compares times series of the individual years from free and initialized CICE simulations with three different CS2 products provided by CPOM, AWI and NASA. For all years, CICE-best-free is much closer to CS2 than CICE-default-free. It is striking that differences

between CICE-best-free and CICE-best-ini are small. This is a remarkable result from a modelling point of view, because it means that CICE-best is so realistic that assimilation of sea ice thickness would not result in substantial changes. This agreement increases the confidence in our new model setup.

Naturally differences between CS2 and CICE-best remain. In November 2012, simulated ice is about 0.5 m thinner than in CS2. During winter 2013/14 the CS2-CPOM ice growth is stronger than in CICE-best-ini with large differences between the

three CS2 products. These differences can be caused either by model errors or by errors estimating sea ice thickness form CS2. As discussed in Section 4.1 using a climatological snow depth limits the applicability to derive inter-annual variability from CS2 ice thickness. For comparison, mean sea ice thickness from PIOMAS (Zhang and Rothrock, 2003) is added in Fig. 8. Although no sea ice thickness observations are assimilated, PIOMAS is widely applied for verification of climate models. In spite of some differences between PIOMAS, CS2 and CICE-best in the sea ice thickness evolution, the general statement that

CICE-best is more realistic than CICE-default is confirmed by PIOMAS.

As the summer sea ice extent is quite realistic in CICE-default (Section 4.2), we investigate the impact of the strong ice thickness increase in CICE-best on sea ice concentration and extent. Fig. 4a reveals that the mean September sea ice extent over the period 2005 to 2014 is slightly underestimated in CICE-default with respect to SSM/I Bootstrap (Comiso, 1999). The impact of the strong thickness increase on sea ice extent is small, the ice extent is only marginally larger in CICE-best (Fig.

4b), but nevertheless close to SSM/I Bootstrap. In contrast to ice extent, ice concentration is underestimated in CICE-default by 25% to 50% in large parts of the ice-covered Arctic Ocean. While apart from the ice edge, ice concentration is generally between 80% and 100% according to SSM/I Bootstrap, CICE-default frequently shows large areas with values below 50%. This discrepancy is strongly reduced in CICE-best with error values below 10% in the Canadian half and between 10% and



30% in the Siberian half of the Arctic (Fig. 4b). It is worth mentioning that nearly all CMIP5 climate models underestimate Arctic summer sea ice concentration in their historical runs with respect to SSM/I Bootstrap (see e.g. Fig. 3 in Notz, 2014). CICE-best does not only improve the sea ice thickness, but also the summer sea ice concentration.

We demonstrate the impact of our model changes on the timing of mean melt and freeze onset (2005-2014) between CICE-best and CICE-default in Fig. 9. In CICE-best the melt onset day is later (0 – 4 days in the Central Arctic, up to 10 days in the Fram Strait) and the freeze onset is earlier (4 – 12 days in most areas) resulting in a shorter melting season. The simulated mean length of the melting season over the Arctic Basin reduces from 107 days (CICE-default) to 100 days (CICE-best). This is an improvement with respect to the observed value of 94 days. The observed number of 94 days is based on a mean value of 88 days for the period 1979 to 2012 and accounting for the trend of 3.7 days/decades (Stroeve et al., 2014). The impact of the model changes is remarkable given we apply the same 2m-air-temperature data (NCEP-2) as atmospheric forcing. Our examples indicate that the chosen model physics may be important for many climate related questions and how climate models predict future changes of summer melting season and sea ice decline.

### 4.5 Impact of initial conditions and atmospheric forcing

How strongly do the CICE simulations depend on sea ice initial conditions? Using the mean CS2 November sea ice thickness from 2010 to 2016 (CICE-climini, Table 3) instead of the annual values leads to positive or negative thickness anomalies which remain throughout the year, becoming only slightly weaker during winter (Fig. 10). The inter-annual variability of the simulated April ice thickness is reduced by 44 % (from 0.18 m to 0.10 m, Table 4) and the September ice thickness by 20 % (from 0.5 m to 0.4 m).

Applying a mean atmospheric forcing for each year (CICE-climforcing-ini) does hardly affect the ice thickness during winter (with the exception of winter 2016/17), but it leads to strong anomalies during summer (Fig. 10). The mean September sea ice thickness is increased by 0.28 m and the inter-annual variability reduced from 0.5 to 0.25 m (Table 4). Applying the mean atmospheric forcing during winter only and the individual atmospheric forcing from May onwards (CICE-climforcing-winter-ini), sea ice thickness during summer is nearly unchanged with respect to CICE-ini. While the atmospheric conditions during summer are decisive for summer ice melt and September sea ice thickness, the atmospheric winter conditions seem not to matter at all.

The atmospheric conditions for the last 8 winters were all relatively warm with respect to the 1980 to 2010 climate. Is it possible that the small variability of these winter periods could cause the weak impact on sea ice thickness? To exclude this possibility, we perform additional sensitivity experiments in which we apply colder atmospheric conditions from the 80's (CICE-80climforcing-ini). Fig. 11 and Table 3 reveal that the impact during winter is nevertheless small (mean thickness increase of 0.11 m), but huge in summer (+0.8 m). Interestingly, the impact of using the wind forcing from an anomalous year (2009/10, stronger transpolar-drift), CICE-climforcing-wind20100ini, is stronger on April ice thickness (-0.13 m) than from the cold atmospheric conditions. This confirms the important role of sea ice dynamics and export through the Fram Strait in controlling the sea ice volume variability (Ricker et al., 2018).

These sensitivity experiments demonstrate that the atmospheric winter conditions have very small impact on winter ice growth and thus on April ice thickness. September ice thickness depends strongly on atmospheric conditions from May to September.

### 5 Conclusions

We determined an optimal region for comparing sea ice thickness between simulations with the sea ice model CICE and CS2 data by taking into account the strengths and weaknesses of both approaches. Since simulating dynamic processes can result in large model errors and can be proportionally less accurate, we exclude locations where sea ice dynamics (sea ice advection and ridging) make a strong contribution in modifying see ice thickness. We further exclude locations where the number of



CS2 observations is limited and the sea ice is thin during winter (< 0.5 m). The resulting region includes most of the Central Arctic, but not the area of the thickest ice north of Canada or any of the shelf seas (see Fig. 3).

Comparing the multi-year means reveals that CICE-default underestimates ice thickness over our comparison region by about 0.8 m (see Fig. 1a). This discrepancy would not have been identified by comparing total Arctic ice volume. Due to
overestimating ice thickness in the Canadian sector, Arctic ice volume from CICE-default is only slightly lower than estimates based on CS2 or PIOMAS. Deriving the sub-grid scale ice thickness distribution (ITD) from CS2 allows us to initialise CICE simulations with the identical ice thickness in November. Applying default settings CICE-ini underestimates mean ice thickness in the following April by 0.25 m (see Fig. 1b). This indicates that the winter ice growth is too weak in the model. What is the reason for the underestimation of ice growth? Our sensitivity experiments give evidence that uncertainty in
atmospheric forcing cannot be the main reason. Impact of errors in air temperature and incoming longwave radiation on sea ice growth is rather small in winter (see Fig. 5). The turbulent ocean-ice heat flux is generally small in winter in the Central Arctic, thus, errors deriving from the turbulent ocean-ice heat flux cannot be responsible either. The ice-atmosphere heat fluxes depend on atmospheric forcing and the transfer coefficients. Varying the transfer coefficients does not result in major changes of sea ice growth. Initial conditions in autumn are important, but our initialized CICE simulations show that April ice thickness
is still underestimated even when starting with the "correct" November ice thickness. Thus, by a process of exclusion, we conclude that sea ice physics related to the conductive flux must be responsible.

The strongest contribution in simulating winter ice growth comes from implementing a snow drift scheme based on Lecomte et al. (2014). Although our implementation is simple and further work to improve the parameterization is required, it demonstrates the importance of including additional snow processes in sea ice models for climate applications (Vionnet et al.,
2012; Nandan et al., 2017; Liston et al., 2018). Such model development can also benefit recent Arctic wide snow products that rely on satellite observations (Maas et al., 2013; Guerreiro et al, 2016; Lawrence et al., 2018) or reanalysis precipitation fields on drifting sea ice (Kwok and Cunningham, 2008).

Our sensitivity experiments modifying initial and atmospheric forcing data reveal that ice thickness anomalies in November decay over winter but are still present in the following April (see Fig. 10). Comparing interannual variability of April ice
thickness between CICE-ini and CICE-climini (Table 4) show that half of the variability comes from the initial conditions. Atmospheric conditions during spring and summer are decisive for summer sea ice conditions, but atmospheric winter conditions have little impact on sea ice growth. Using "cold" forcing from the 80's instead of the more recent winters leads to an increase in September sea ice thickness of 0.8 m, but only to an increase in April ice thickness of 0.11 m (see Fig. 11 and Table 4). This reflects the importance of feedback processes. During winter, the negative conductive feedback process (less
ice, more growth) is dominating, while during summer the positive albedo feedback process determines sea ice changes. The impact of the negative winter feedback has been discussed in Stroeve et al. (2018). In their study a potential weakening of the feedback during the last years has been raised as a question. Here, we answer this question demonstrating that warm winters are not important for observed sea ice thinning in the last decades, at least in the Central Arctic. The situation can be different in the marginal winter sea ice zone, where a warm winter can increase mixed-layer ocean temperature and delay ice growth.
Our findings are in agreement with observations of the last years: In spite of the three warmest Arctic-wide winter air temperatures during 2014/2015, 2015/2016 and 2016/2017 on record (Stroeve et al., 2018), the September Arctic sea ice extent in 2015 (4.7 million km²), 2016 (4.7 million km²) and 2017 (4.9 million km²) has been larger than in 2012 (3.6 million km²), numbers based on SSM/I NASA-Team algorithm (Cavalieri et al. 1996, updated 2017). We conclude that the fate of Arctic summer sea ice depends largely on atmospheric spring and summer conditions, in particular May and June, when the melting
season starts and melt ponds form, preconditioning the strength of the positive albedo feedback mechanism (Schröder et al., 2014).

Our optimal model configuration CICE-best does not only improve the simulation of mean sea ice thickness over the Central Arctic with respect to CS2, but also improves summer sea ice concentration (in comparison to SSM/I Bootstrap), the length





of melt season (in comparison to Stroeve et al., 2014) and melt pond fraction (in comparison to MODIS and MERIS). Recent studies demonstrated improvements for sea ice predictions up to 6 months initializing forecast models with CS-2 data (Allard et al., 2018; Blockley and Peterson, 2018). Here, we show that our improvements to the sea ice model CICE are so fundamental and consistent that any differences between CICE simulations initialized by CS2 ice thickness and those which do not utilize
them are minimized. It is the first time CS2 sea ice thickness data have been applied successfully to improve sea ice model physics.

*Acknowledgements.* This work was done under the ACSIS and UKESM program funded by the U.K. Natural Environment Research Council. The authors would like to thank the Isaac Newton Institute for Mathematical Sciences for support and
hospitality during the programme Mathematics of sea ice phenomena when work on this paper was undertaken. This work was supported by EPSRC grant number EP/K032208/1". MT acknowledges the support from the Arctic+ European Space Agency snow project ESA/AO/1-8377/15/I-NB NB - 'STSE - Arctic+.

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

| Run Name | Description | free | ini |
|---|---|---|---|
| CICE-mw | The maximum **m**elt **w**ater added to melt ponds rfracmax is reduced from 100% to 50%. The actual fraction of melt water rfrac is calculated as: rfrac = rfracmin + (rfracmax-rfracmin) * aice with rfracmin = 15%. This reduction accounts for the uncertainty in the fraction of melt water that collects in ponds versus being immediately released to the ocean. The impact of this change on the simulated melt pond fraction in July 2012 is demonstrated in Fig. 1. The melt pond fraction in the central Arctic is generally reduced by 5-10% ranging from 25 to 40% in the default simulation and from 20 to 35% in CICE mw. The new melt pond distribution is more realistic with respect to MODIS derived melt pond fractions (Roesel et al., 2012). | Y | Y |
| CICE-mw-form | Instead of a constant drag coefficient for the momentum fluxes between atmosphere and ice (CDa = 1.3 x 10-3) and between ice and ocean (CDo = 5.36 x 10-3), the **form** drag parametrization of Tsamados et al. (2014) is applied accounting for the impact of pressure ridges, keels, ice floe and melt pond edges. Here, we modify the background drag coefficient for the atmosphere (csa = 0.01 instead of 0.005) and the ocean (csw = 0.0005 instead of 0.02) and the parameters determining the impact of ridges and keels (cra = 0.1 instead of 0.2 and crw = 0.5 instead of 0.2). These modifications increase ice drift over level ice and decrease ice drift over ridged ice resulting in a more realistic ice drift pattern in comparison to Pathfinder (not shown). | Y | Y |
| CICE-mw-form-e | The longwave **e**missivity of sea ice is increased from 0.95 to 0.976. | Y | Y |
| CICE-mw-form-e-sd | Depending on wind speed, snow density and surfaceaography, **s**now can be eroded from the sea ice surface, **d**rift through air and be redistributed or lost in leads. The default CICE simulation does not account for these processes. Here, we parameterize the snow erosion rate following Lecomte et al. (2014): $$\frac{\partial h_s}{\partial t} = -\frac{\gamma}{\sigma_{ITD}}(V - V^*)\frac{\rho_{s,MAX} - \rho_s}{\rho_s}$$ with snow depth $h_s$, mass flux tuning coefficient $\gamma = 10^{-5}$ kg m$^{-2}$, current wind speed $V$, threshold wind speed $V^* = 3.5$ m s$^{-1}$, current snow density $\rho_s$ and maximum snow density $\rho_{s,MAX}$, and standard deviation of ice thickness distribution $\sigma_{ITD}$. Lacking information about the snow density distribution, we apply $\rho_{s,MAX} = 330$ kg m$^3$ (the constant snow density in CICE) and assume $\rho_s = 240$ kg m$^3$. Regarding the ITD, we apply $\sigma$ values of 0.25 m for ice category 1 (ice thickness $h < 0.6$ m), 0.5 m for category 2 (0.6 m $< h <$ 1.4 m) and 1 m for category 3 1.4 m $< h <$ 2.4 m). We assume that the whole amount of snow blown into the air will be released into the ocean. Estimating the error of this assumption, we calculate the net snow re-deposition rate. Snow which is blown into air, will be deposited at the surface and might be blown into the air again if the wind speed stays above the threshold value. Assuming an average friction velocity of 0.1 m s$^{-1}$ and a total distance of 200 m, one cycle takes approximately 30 min. For every cycle, the lead fraction defines the fraction of snow volume released into the ocean. Analyzing NCEP-2 wind fields, the average period the wind speed stays above the threshold value of 3.5 m s$^{-2}$ ranges from 50 to 120 h over the Arctic sea ice with the lowest values close to coast of North Greenland. Thus, for most parts of the Arctic more than 90 % of the total snow blown into the air would be lost in leads. An error of less than 10 % justifies our simplification. The impact of our parameterization on the simulated snow depth can be seen in Fig. 2. Accounting for the loss of drifting snow reduces the snow depth between 20 and 40 %. The larger differences occur over regions with strongest winds and the smallest differences north of Greenland and Canada. | Y | Y |
| CICE-mw-form-e-sd-bubbly (CICE-best) | We apply the **bubbly** conductivity formulation from Pringle et al. (2007) which results in larger thermal conductivity values for colder ice temperatures. | Y | Y |

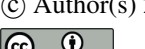



712.        **Table 2**. Sensitivity simulations exploring the impact of uncertainty in atmospheric forcing data. "Free" indicates multi-year simulations from 1980 to 2017; "ini" indicates seven 1-year-long simulations starting in mid-November with CS2 sea ice thickness (2010/2011 to 2016/17).

| Run Name | Description | free | ini |
|---|---|---|---|
| CICE-Ldown15 | As CICE-default, but forcing field incoming longwave radiation has been decreased by 15 % everywhere and for all times. | N | Y |
| CICE-Tair2 | As CICE-default, but forcing field 2m-air temperature has been decreased by 2 K everywhere and for all times. | N | Y |

712.        **Table 3**. Sensitivity simulations exploring the impact of initial and forcing conditions. "Free" indicates multi-year simulations from 1980 to 2017; "ini" indicates seven 1-year-long simulations starting in mid-November with CS2 sea ice thickness (2010/2011 to 2016/17). Model configuration for all simulation is CICE-best.

| Run Name | Description | free | ini |
|---|---|---|---|
| CICE-climini | The same climatological initial conditions from CS2 CPOM (mean over 2010-2016) are applied for each November. | N | Y |
| CICE-climforcing | Climatological atmospheric forcing calculated over the period 2011 to 2017 is applied for air temperature, humidity, downward longwave and shortwave radiation, rainfall and snowfall. The real wind forcing is applied in all simulations. | N | Y |
| CICE-climforcing-winter | Climatological atmospheric forcing is applied from mid-November to April and real forcing afterwards (May to November). | N | Y |
| CICE-80climforcing | As CICE-climforcing, but climatological atmospheric forcing calculated over the period 1981 to 1987. | N | Y |
| CICE-80climforcing-winter | As CICE-climforcing winter, but climatological atmospheric forcing calculated over the period 1981 to 1987. | N | Y |
| CICE-climforcing-wind2010 | As CICE-climforcing, but using the wind forcing from 2009/10 instead of the real wind forcing. | N | Y |

712.        **Table 4**. Impact of initial November sea ice conditions and atmospheric forcing on mean sea ice thickness of subsequent April and September over region shown in Fig. 3 from 2011 to 2017. See Section 3.3 for explanation of CICE simulations. Inter-annual variability is given in parenthesis.

| Setup | April-hi in m | September-hi in m |
|---|---|---|
| CICE-ini | 2.70 (+/- 0.18) | 1.62 (+/- 0.50) |
| CICE-clim-ini | 2.67 (+/- 0.10) | 1.57 (+/- 0.40) |
| CICE-climforcing-ini | 2.70 (+/- 0.17) | 1.90 (+/- 0.25) |
| CICE-climforcing-winter-ini | 2.70 (+/- 0.17) | 1.62 (+/- 0.50) |
| CICE-80climforcing-ini | 2.81 (+/- 0.16) | 2.37 (+/- 0.21) |
| CICE-80climforcing-winter-ini | 2.81 (+/- 0.16) | 1.78 (+/- 0.49) |
| CICE-climforcing-wind2010-ini | 2.57 (+/- 0.17) | 1.92 (+/- 0.17) |





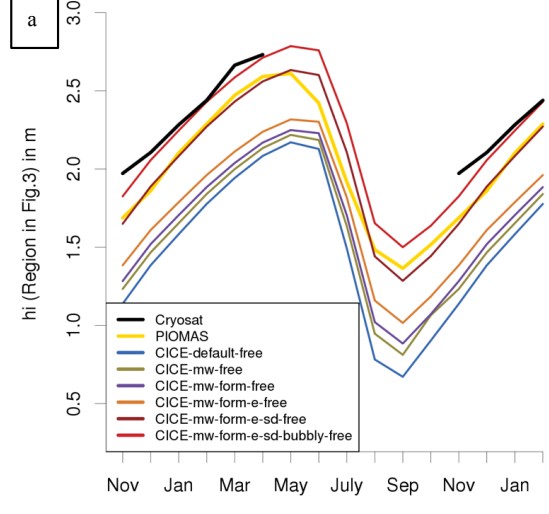

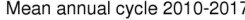

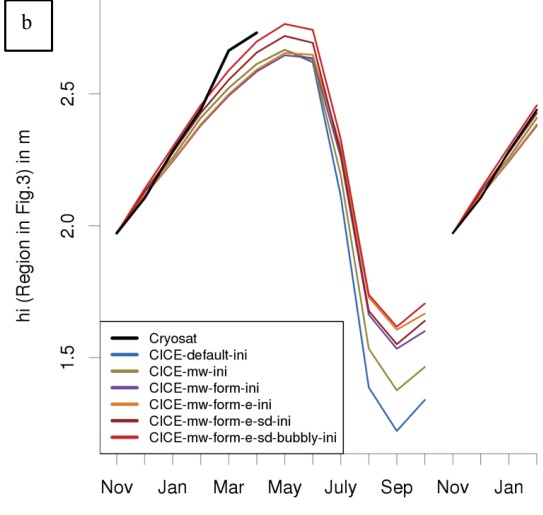

**Figure 1: Impact of CICE setup on mean effective sea ice thickness over region shown in Fig. 3 and averaged over winter periods 2010/2011 to 2016/2017: a) CICE-free simulations (multi-year runs from 1980 to 2017) and b) CICE-ini simulations (seven 1-year-long runs starting in mid-November with CS2 sea ice thickness). Model results are compared with mean sea ice thickness from CS2 CPOM and PIOMAS. Effective ice thickness (divided by ice concentration) is presented. For CS2 ice concentration is applied from CICE-default (mean values for November to April vary between 99.4 and 998. %. See Section 3.3.1 for explanation of sensitivity experiments.**

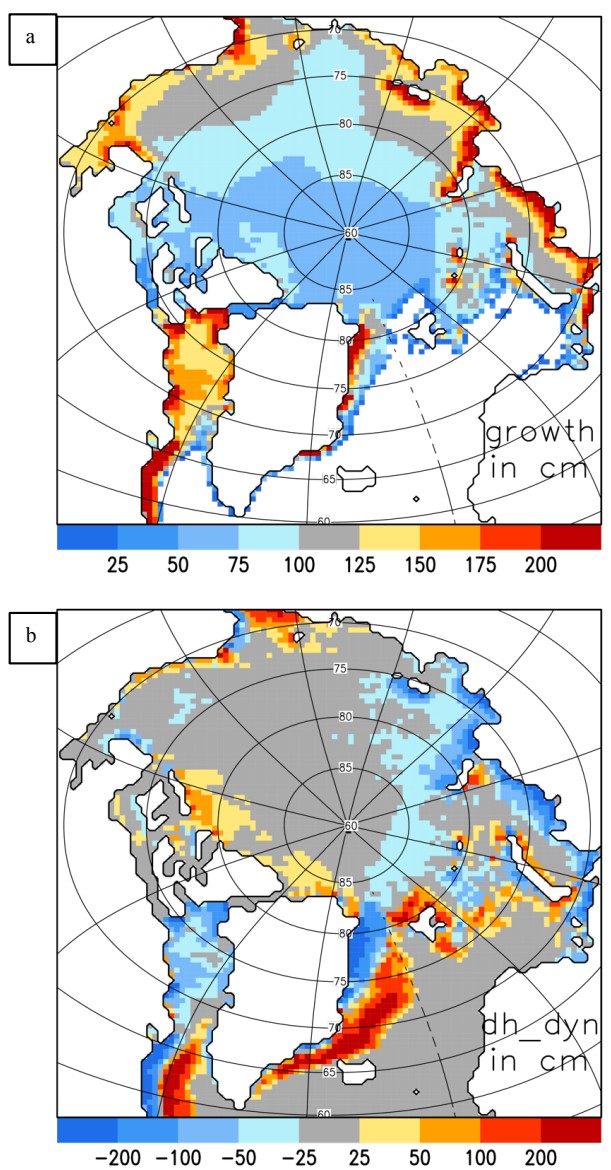

**Figure 2:** **Mean simulated sea ice thickness change from November to April (2010-2016) in CICE default: a) Sea ice growth in cm and b) Sea ice thickness change by dynamical processes dh_dyn in cm (advection, convergence and ridging).**





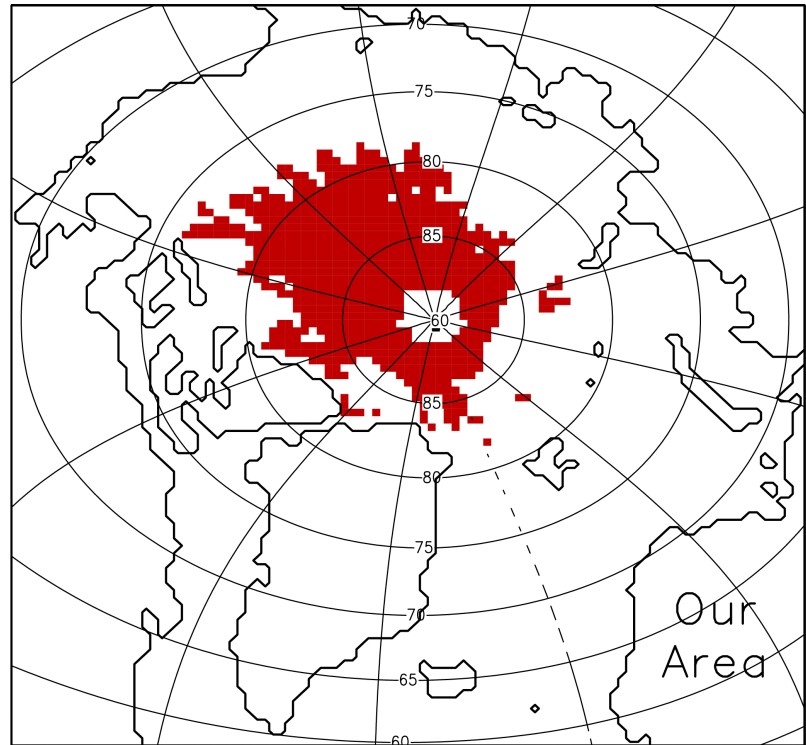

**Figure 3:** Region for comparing CICE and CS2 sea ice thickness where impact of winter growth on thickness change dominates other factors and where CS2 data are most accurate (See Section 4.1 for more details).



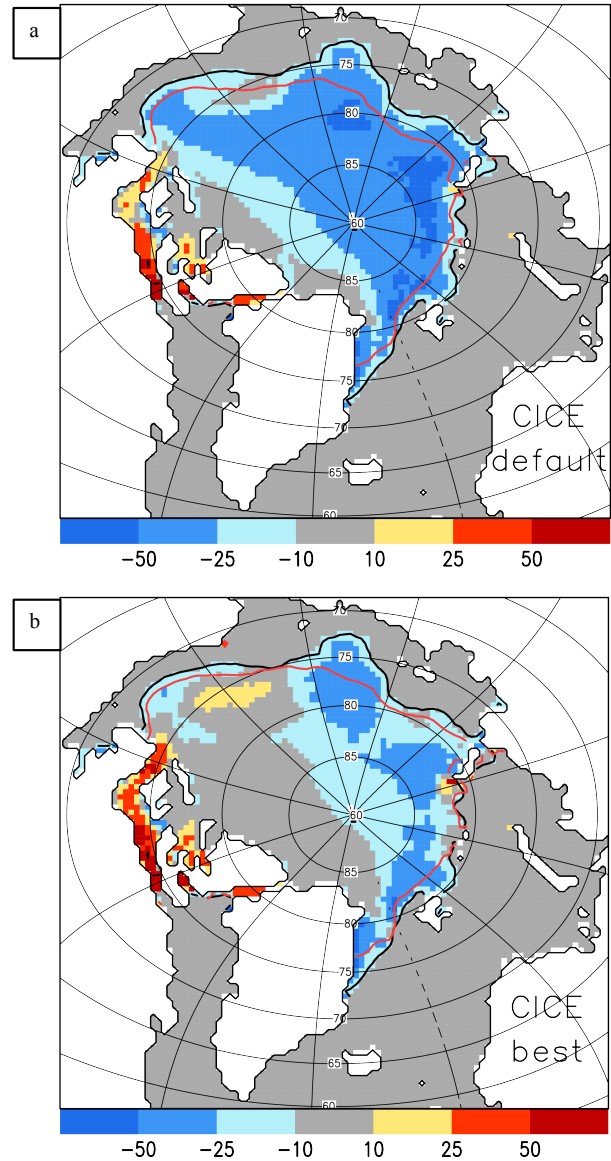

**Figure 4: Difference in mean September sea ice concentration (2005-2014) in % between CICE simulation and SSM/I Bootstrap data. Negative values mean lower ice concentration in CICE. Black line indicates mean SSM/I and red line mean CICE sea ice extent: a) CICE-default and b) CICE-best.**





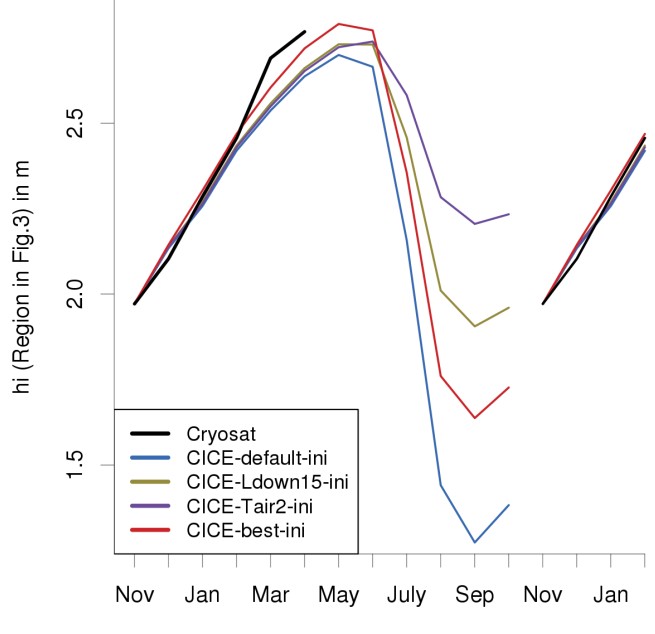

**Figure 5: Impact of uncertainty in atmospheric forcing on simulated mean effective sea ice thickness over region shown in Fig. 3 and averaged over winter periods 2010/2011 to 2016/2017. CICE-ini simulations: default, default with increase of incoming longwave radiation of 15 Wm$^{-2}$, default with increase of 2m air temperature of 2 K (everywhere and anytime), best. Model results are compared with mean sea ice thickness from CS2 CPOM.**





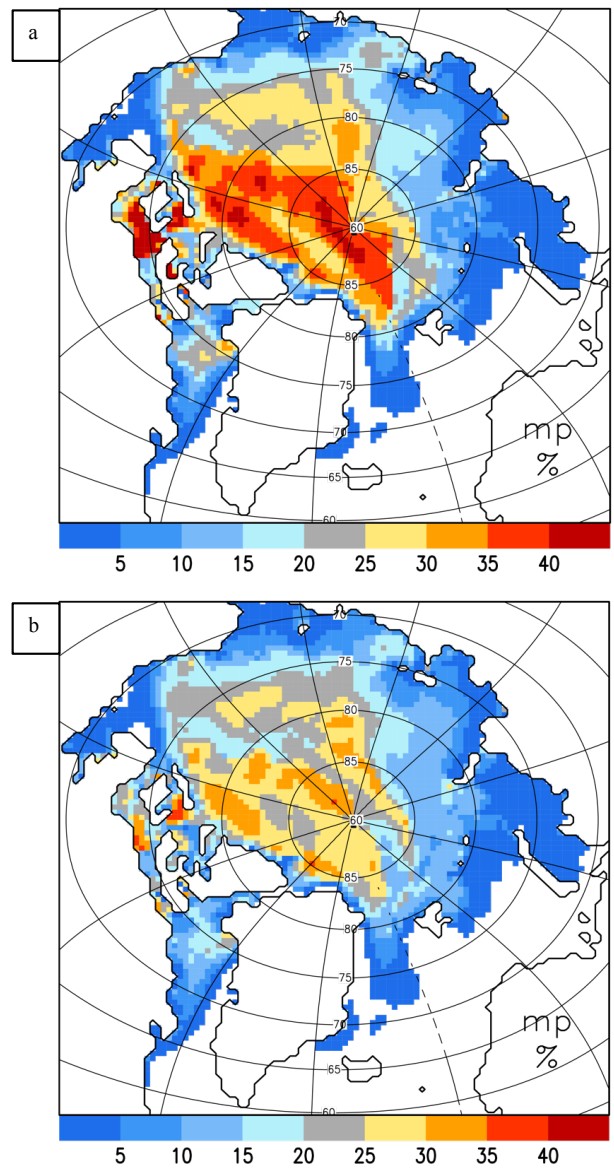

**Figure 6:** **Impact of reduced retained melt water fraction on simulated melt pond area fraction in July 2012: a) CICE-default and b) CICE-mw.**



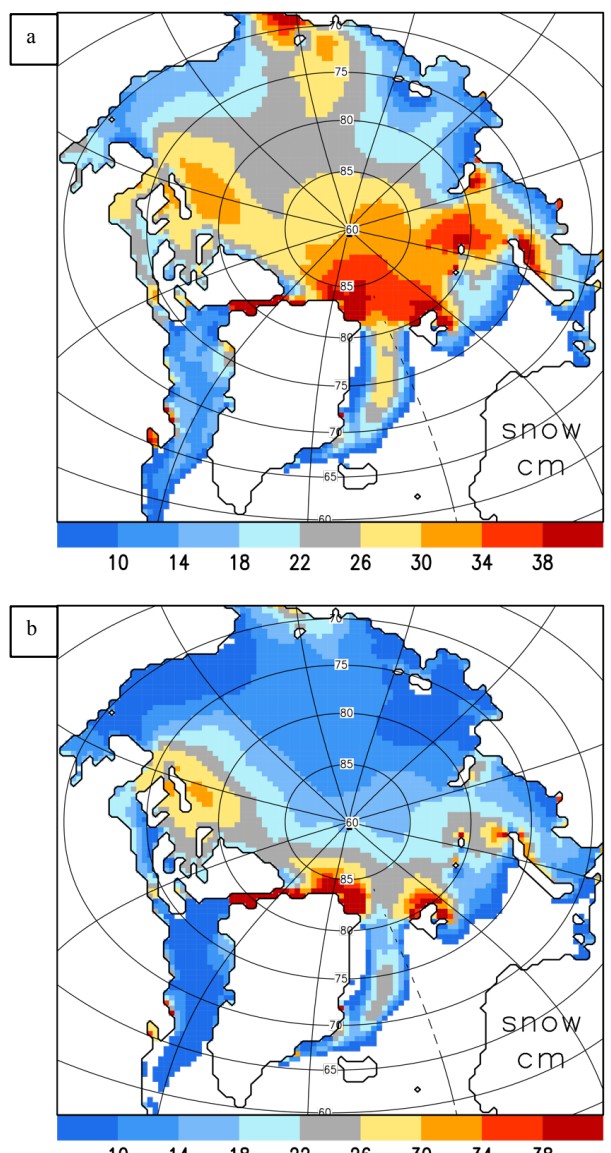

**Figure 7: Impact of snow erosion on simulated mean April snow depth (2005 to 2014): a) CICE-default and b) CICE-mw-from-e-sd.**





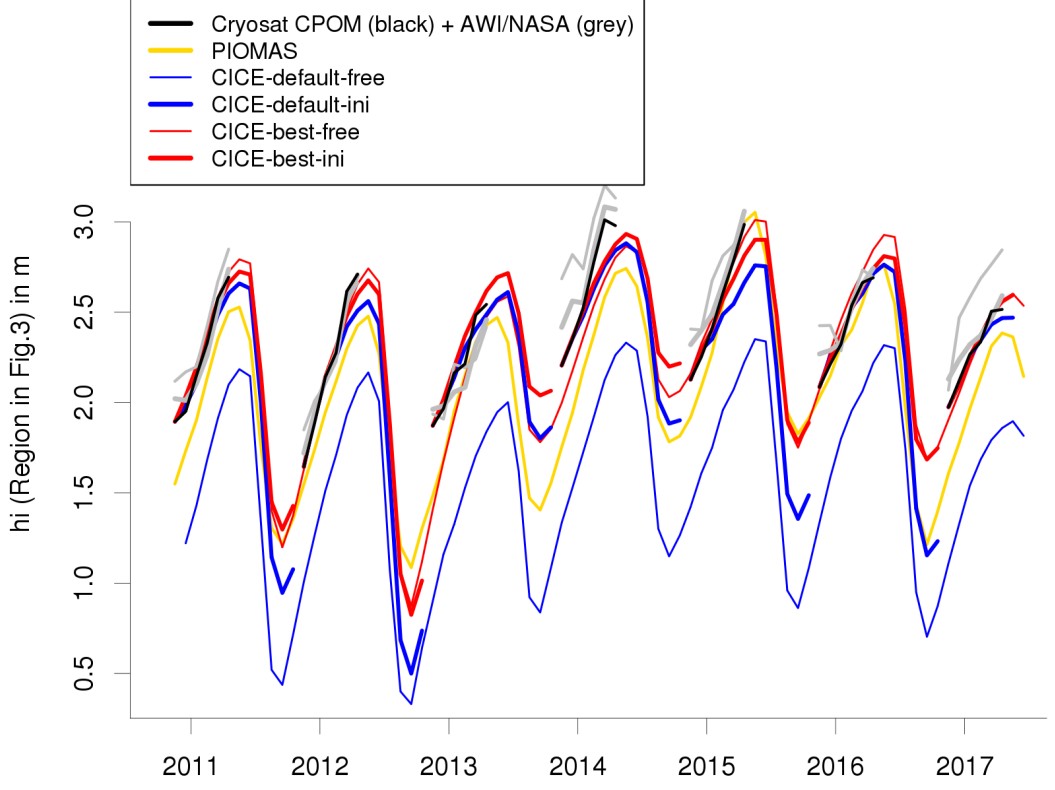

**Figure 8: Mean effective sea ice thickness over region shown in Fig. 3. Verification of CICE-free default and best (thin lines) and CICE-ini default and best (thick lines) with CS2 CPOM, AWI and NASA and PIOMAS.**





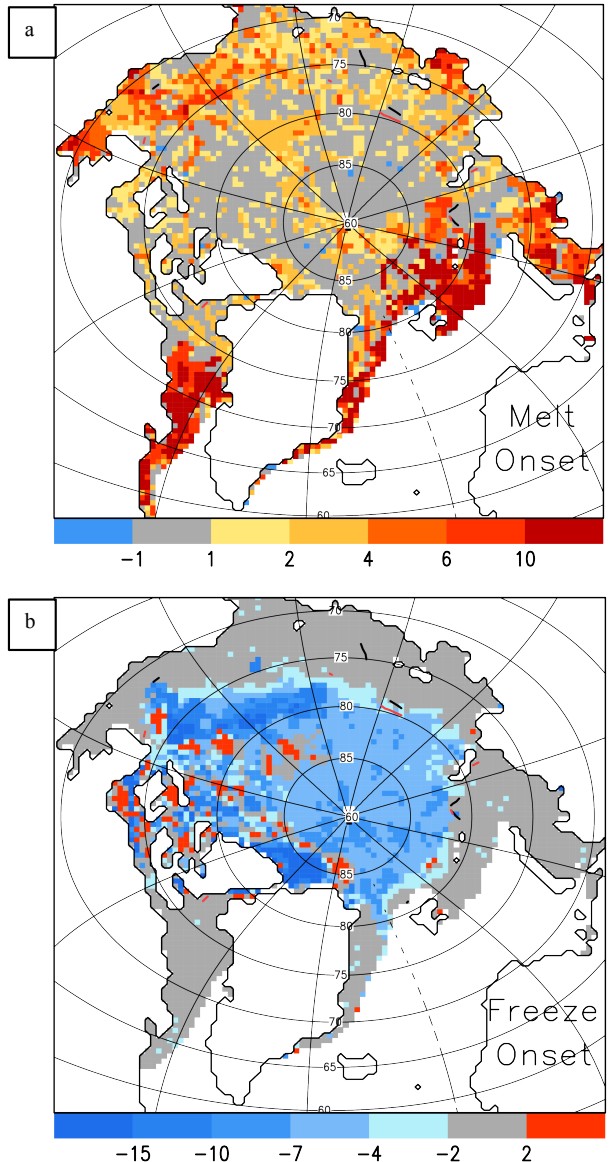

**Figure 9: Difference in mean melt onset (a) and freeze onset (b) in days between CICE-best and CICE-default (2005-2014). Positive values mean later onset day in CICE-best.**





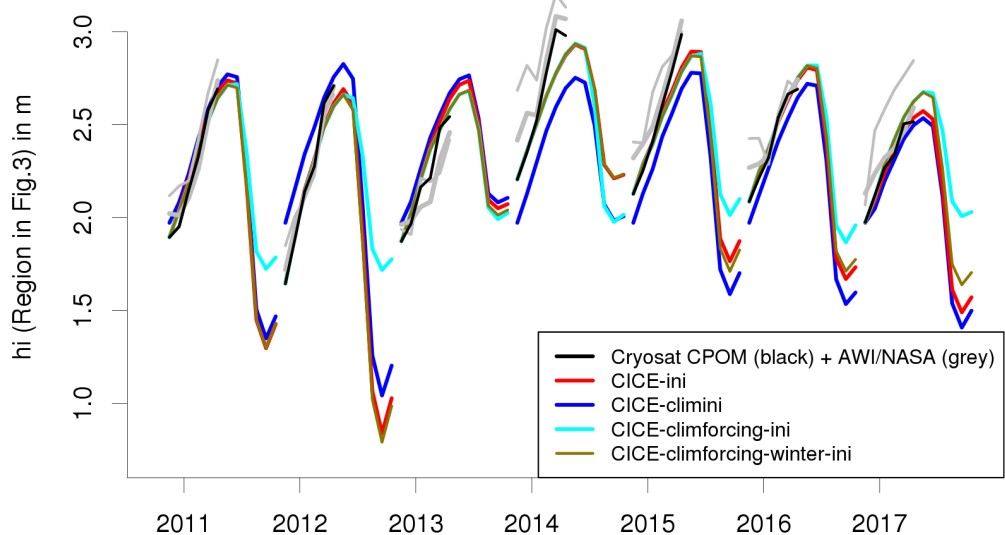

**Figure 10: Impact of climatological initial conditions (climini), climatological forcing (climforcing) and climatological forcing during winter only (November to April, climforcing winter). All CICE simulations from CICE-ini best. Mean effective sea ice thickness over region shown in Fig. 3. CS2 as in Fig. 7. The climatology is calculated over the period 2011 to 2017 for air temperature, humidity, downward longwave and shortwave radiation, rainfall and snowfall. The real wind forcing is applied in all simulations.**





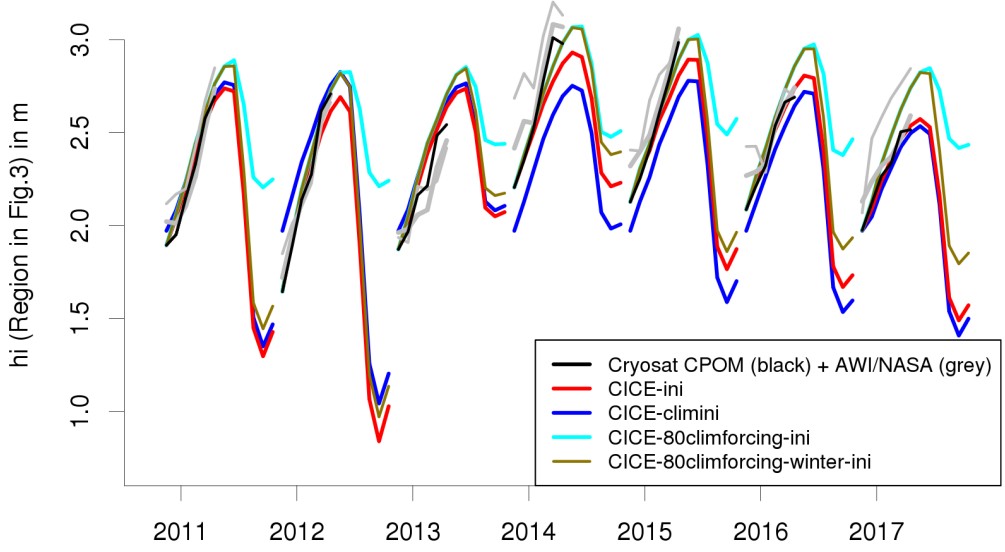

Figure 11: As Fig. 10, but applying a climatology calculated over the period 1981 to 1987.