# Peer review of "New insight from CryoSat-2 sea ice thickness for sea ice modelling"

_The Cryosphere, 2018_

## Referee Comment (RC1) · Anonymous Referee #1 · 4 Sep 2018

General comments:

GC1: The authors assume that CS2 estimates are unbiased towards thicker sea ice and superior to CICE in general, and that there is a negative sea ice thickness bias in CICE estimates. Given the substantial uncertainties in CS2 estimates, it would be better to phrase comparisons with ambiguity regarding which is more accurate: e.g. "Comparing the CICE simulation with CS2 reveals that CICE default underestimates the mean monthly sea ice thickness" would become "Comparing the CICE simulation with CS2 reveals that CICE default produces thinner mean monthly sea ice". Furthermore, given the importance of CS2 estimates, its uncertainty and bias should be quantitatively described, early in the paper, for each of the five sea ice thickness classes used. As shown in Tilling et al. (2018), CS2 biases for ice < 1.4 m are generally positive, whereas for ice > 3.6 m biases are generally negative; and Ricker et al. (2017) provide an assessment of relative error by thickness. The uncertainties and biases, as a factor of ice thickness, should be more prominent in discussions of the results.

GC2: The region (grid cells) chosen for CICE - CS2 comparison is an area usually dominated by multiyear sea ice. This is evident in Figure 1 where the ice does not melt fully during the summer, indicating primarily multiyear sea ice. This is also evident in that the September model results for the grid cells show a mean ice thickness > 1.57 m (Table 4). Please describe the proportions of multiyear versus first-year sea ice that make up the comparison grid cells. If first-year sea ice constitutes a significant proportion, the vertical shift in the scattering horizon due to snow salinity (Nandan et al., 2017) could account for most of the 0.8 m CICE - CS2 difference observed. However, for multiyear sea ice with its non-saline snow this would not apply. If the analysis is primarily limited to multiyear sea ice, this should be reflected throughout the paper. Please include a figure showing the frequency distributions of ice thickness in November and in April.

GC3: The authors attempt to force the model to fit the CS2 estimates by making cumulative changes to the model that are biased towards thickening the sea ice. While these are reasonable for assessing which variables may be incorrectly parameterized, they are not really sensitivity studies; they are rather specific model perturbations intended to produce thicker sea ice. A terminology other than "sensitivity study" should be used.

GC4: When comparing the CICE-mw-form-e-sd and subsequent models with CS2 estimates, the snow distributions will be different, assuming W99 is used for CS2 estimates. W99 for the comparison grid cells in April is approximately 30 cm, whereas Figure 7b suggest values of half that thickness; the resulting CS2 sea ice thickness estimates would be substantially thicker. If the substantially thinner snow cover distribution (as per Figure 7b) were to be used for the CS2 estimates, then a direct comparison would be more valid.

Specific comments:

P2, L3: Please include a short description of CICE, with references.

P2, L20: Discrimination of sea ice type is also a principle challenge (in order to assign ice densities and snow loads); especially when the effects of snow salinity for first-year sea ice are also considered.

P2, L32: Can it be called a "full ice thickness distribution" if values < 0.5 m are not included?

P2, L35: Given that "A realistic ITD is critical for simulating ice growth and ice melt rates correctly", an explanation should be included here on the impact of omitting thin ice points.

P3, L31: In order to compare the uncertainties of CS2 estimates with the uncertainties of CICE model results, please summarize previously reported (if any) uncertainties and biases, for sea ice thickness, of CICE-default.

P4, L3: See GC3.

P4, L25: Multiyear sea ice is mentioned, but please include a statement regarding the accuracy of snow on first-year sea ice.

P4, L26: In "Therefore, we apply a climatology for the CS2 period 2010 to 2017", what is meant by "a climatology"? Is this W99 as per P2, L23?

P5, L10: The language used is too strong: e.g. "This indicates that the winter ice growth is underestimated in the model." See GC1. Recommend: "This suggests that the winter ice growth may be underestimated in the model."

P5, L10: If the CS2 ice thickness is overestimated when thinner and more-accurately estimated (or underestimated) when thicker (as per Tilling et al., 2018) then this may mitigate the CICE April "underestimation".

[Figure]

P5, L29: This statement is rather broad, given that only air temperature and incoming longwave were assessed, and then only for two specific perturbations. Please reword.

P6, L11 and Table 1: For CICE-mw-form-e, what is the justification for increasing the longwave emissivity of sea ice from 0.95 to 0.976.

Technical corrections:

Figure 1 caption: "(mean values for November to April vary between 99.4 and 998. %." Is missing a ")" and 998 is likely 99.8%.

Figure 3: Change North pole latitude to 90, or omit. In the caption, would not "grid cells" be more accurate than "region".

P3, L26: Define "SIT".

Table 1 caption: "...model changes..."

Table 1: surfaceaography = surface topography?

---

## Referee Comment (RC2) · Anonymous Referee #2 · 2 Oct 2018

General Comments

In this paper, the authors evaluate the ice thickness from a standalone CICE model forced with NCEP-2 Reanalysis atmospheric forcing against CryoSat-2-derived ice thickness during 2010-2017 to determine the cause of the model's underestimation of the central Arctic winter ice thickness. A multi-year simulation is performed (CICE-free) for the period of 1980-2017 on a 40 km CICE (v5.1.2) grid using a simple mixed layer ocean model where the ocean temperature is restored to climatology on 20-day restoring time scale and utilizes monthly mean climatological ocean currents. Model defaults include a prognostic melt pond, elastic anisotropic rheology, ridging and the Delta-Eddington radiation scheme. Comparisons against CryoSat-2 (CS2) sea ice thickness from CPOM, AWI, and NASA for the months of November – April all reveal a significant

under-prediction in the winter Central Arctic ice thickness. The CS2 data is binned into five CICE model ice thickness categories ranging from the first category with < 0.6 m of ice to the fifth category with ice > 3.6 m on a rectangular 50 km grid and used to initialize a series of experiments. The paper discusses the treatment of snow for each CS2 dataset in deriving the ice thickness. In their study, the authors chose a region in the Central Arctic (Fig. 3) with a minimum ice thickness of 0.5 m and a region where ice growth is dominant over the impact of sea ice dynamics. A series of 7 studies are performed for the multi-year periods beginning in November of 2010 and continuing for 15 months and repeated through 2016/2017. Experiment are initialized with all three CS2 ice thickness fields and use different CICE options documented in Table 1.

An experiment was performed by decreasing the longwave radiation by 15 w/m2 and reducing the air temperature by 2 K everywhere and for all times. They found some differences in the mean September ice thickness, but little change in winter ice thickness; concluding that the under-prediction in winter ice growth can not be explained by errors in forcing data.

Experiments were conducted which studied the impact of increased total ice surface albedo by releasing more melt water into ocean; thus reducing ponded water over the surface of ice. This study (CICE-mw) only revealed a small improvement in winter ice thickness.

Another set of experiments applied the form drag parameterization of Tsamados et al. (2014) to investigate sea ice advection and turbulent heat fluxes (CICE-mw-form). They found a small ice thickness increase attributed to a reduce ice drift speed resulting in a weaker ocean-ice heat flux. Another set of experiments increased the emissivity of snow from 0.95 to 0.976 (CICE-mw-form-e) impacting the longwave radiation budget. They found that the summer melt was reduced by only a few cm, but no noticeable change was found in winter.

The most significant improvement was found with the implementation of a snow drift

scheme based on Lecomte et al. (2014) which reduced the snow depth by 20-40% (CICE-mw-form-e-sd) which lowered the winter ice thickness errors from 0.8 to 0.25 m. They found that the reduction of snow led to larger winter ice growth as well as an increase in the summer melt due to the earlier disappearance of snow. A final set of studies (CICE-mw-form-e-sd-bubbly) increased the conductivity for colder temperatures (Pringle et al., 2007) and further reduced the errors to less than 0.1m. This configuration is referred to as "CICE-best".

Using the CICE-best configuration, the authors studied the length of the melt season and found a 7-day reduction from 107 to 100 days versus the "observed" value of 94 days. They performed studies using "cold forcing from the 1980's" which led to an increase in the September ice thickness of 0.8 m, but only 0.11 m in April.

The authors conclude that warm winters are not responsible for the observed sea ice thinning in the Central Arctic over the past decade; rather it depends strongly on atmospheric conditions from May to June in particular, with the start of the melt season and formation of melt ponds, thus preconditioning the strength of the positive albedo feedback. They found that the recommended improvements to the CICE model outweigh the impact of initialization with CS2 ice thickness fields.

This is a very well written and thorough paper with carefully designed model experiments. I highly recommend it for publication.

Specific Comments

Throughout the paper your refer to 1-year experiments with the CS2 data, but your time series plots look like the experiments run from (for example) Nov 2010 – January 2012, so a 15-month experiment?

The CICE tripole grid has a resolution of ∼40 km. Why wasn't the CS2 data interpolated onto that same 40km grid (instead of 50 km)?

Please review the use of colors chosen for Figures 1, 5, 8, 10, and 11. Some colors are

a bit difficult to differentiate. Perhaps including dashed lines and reducing the number of colors could help.

Technical Corrections

Page 2 line 2: replace "Blockley et al" to "Blockley and Peterson"

Page 2 line 28: Spell out ORCA

Page 2 lines 33,34: replace "Libscomp" to "Lipscomb"

Page 5 line 20: add "by 2 K during the whole simulation. . ."

Page 6 line 2: replace "satellite products" to "satellite products (not shown)"

Page 8 line 21: should be "Maaß"

Page 13 Table 1 caption: should be "Note that all model changes. . ."

Page 15 Figure 1 caption: replace "998." With "99.8%"

Page 22: Why are AWI/NASA both showing up as solid grey lines? Can one be shown as dashed?

Page 24, 25: same question as above

---

## Author Comment (AC1) · 9 Nov 2018

Please see uploaded PDF with our response and suggested changes in manuscript as Supplement file.

Please also note the supplement to this comment:
https://www.the-cryosphere-discuss.net/tc-2018-159/tc-2018-159-AC1-supplement.pdf

---

## Author Response (AR1)

**Response to referee comments**

(Referee comments are shown in black, our response in blue and changes in manuscript in red. See also attached revised manuscript with highlighted changes.)

**Anonymous Referee #1**

**Response: First, we would like to thank the reviewer for the time spent on our study and the thoughtful comments.**

**General comments:**

GC1: The authors assume that CS2 estimates are unbiased towards thicker sea ice and superior to CICE in general, and that there is a negative sea ice thickness bias in CICE estimates. Given the substantial uncertainties in CS2 estimates, it would be better to phrase comparisons with ambiguity regarding which is more accurate: e.g. "Comparing the CICE simulation with CS2 reveals that CICE default underestimates the mean monthly sea ice thickness" would become "Comparing the CICE simulation with CS2 reveals that CICE default underestimates the default produces thinner mean monthly sea ice". Furthermore, given the importance of CS2 estimates, its uncertainty and bias should be quantitatively described, early in the paper, for each of the five sea ice thickness classes used. As shown in Tilling et al. (2018), CS2 biases for ice < 1.4 m are generally positive, whereas for ice > 3.6 m biases are generally negative; and Ricker et al. (2017) provide an assessment of relative error by thickness. The uncertainties and biases, as a factor of ice thickness, should be more prominent in discussions of the results.

Response: In Tilling et al. (2018) CS-2 thicknesses were compared with ice thickness from three different independent datasets and they concluded that there is no significant bias between satellite data and in situ measurements. The absolute differences between CS2 derived thickness and independent observations may arise through uncertainties in either data sets, as the in-situ measurements are still a derived product with their own associated uncertainties. Therefore, it is not possible to say what dataset causes the slight variation in bias with ice thickness, and we cannot assume it is CS2.

Ricker et al (2017) showed that sea ice thickness uncertainty increases over thin ice, with a rapid increase at ~0.5 m. However, these thicknesses are not included in our analysis, as already stated in the manuscript (Section 2). Tilling et al (2018) quote a general grid cell uncertainty of 25%.

Based on the assessment of the relative error depending on ice thickness by Ricker et al. (2017), we decided not to include grid cells with a mean ice thickness below 0.5 m. We discuss uncertainties of CS2 estimates in Section 2 and Section 4.1 in detail and provide numbers for impact of error of assumed snow depth. Our selected region and the mean over the CS2 period are aimed to reduce the impact of these uncertainties. The difference in monthly mean ice thickness over our selected region and over the CS2 period of 0.8 m between CS2 and CICE-default cannot be explained by CS2 uncertainties. Therefore, the differences must be caused to a large extent by model errors. We are not comparing two different thickness products with similar accuracy. CS2 thickness estimates are more accurate than CICE-default for our comparison.

**Suggested change in manuscript: We add the 25% value in grid cell uncertainty when introducing CPOM data in Section 2.**

GC2: The region (grid cells) chosen for CICE - CS2 comparison is an area usually dominated by multiyear sea ice. This is evident in Figure 1 where the ice does not melt fully during the summer, indicating primarily multiyear sea ice. This is also evident in that the September model results for the grid cells show a mean ice thickness > 1.57 m (Table 4). Please describe the proportions of multiyear versus first-year sea ice that make up the comparison grid cells. If first-year sea ice constitutes a significant proportion, the vertical shift in the scattering horizon due to snow salinity (Nandan et al., 2017) could account for most of the 0.8 m CICE - CS2 difference observed. However, for multiyear sea ice with its non-saline snow this would not apply. If the

analysis is primarily limited to multiyear sea ice, this should be reflected throughout the paper. Please include a figure showing the frequency distributions of ice thickness in November and in April.

Response: The field data used in the Nandan et al (2017) study were collected over the late winter season (April and May) in the Canadian Archipelago. We would expect the saline properties of snow on sea ice to differ throughout the season and with location, even just over FYI. Therefore, we do not wish to assume that the impact of salinity found in the study is applicable Arctic-wide, and indeed across our whole study region. In any case, Nandan et al (2017) found the impact of salinity was to shift the scattering horizon by 0.07 m. This would lead to an overestimate in ice thickness of about 35 cm for a typical FYI freeboard and snow thickness. Our selected region combines multi-year ice and first-year sea ice. The mean first-year fraction in April amounts to 35 %, so the mean bias caused by salinity would amount to 0.35 x 0.35 m = 0.1225 cm – clearly less than 0.8 m. Thus, we can exclude that snow salinity effect could account for a large part of the 0.8 m CICE-default - CS2 difference.

Suggested change in manuscript: We add statement about the impact of salinity and the ratio first-year vs multi-year ice for our selected region in Section 4.1 and 4.4. We also add a new table showing the area fraction for each of the 5 ice thickness categories. In November 25 % of the region are covered by ice which is thinner than 1.4 m and 4 % by ice which is thicker than 3.6 m according to CS2. In April the thin ice fraction reduced to 12 % and thick ice fraction increased to 22 %. While the mean ice thickness between CICE-best-ini and CS2 is very similar, the thickness distribution is a bit narrower in CICE in April: 6 % of area thinner than 1.4 m and 10 % thicker than 3.6 m.

Table 4: Mean area fraction of sea ice per category according to CS2 and CICE-best-ini (over region shown in Fig. 3 and period 2010 to 2017).

| Ice thickness (h)            | CS2 | CS2   | CICE-best-ini |
|------------------------------|-----|-------|---------------|
| category                     | Nov | April | April         |
| 1 ( h < 0.6 m)        | 8   | 5     | 2             |
| 2 (0.6 m < h < 1.4 m) | 17  | 7     | 4             |
| 3 (1.4 m < h < 2.4 m) | 43  | 23    | 30            |
| 4 (2.4 m < h < 3.6 m) | 27  | 42    | 54            |
| 5 ( h > 3.6 m)        | 4   | 22    | 10            |

GC3: The authors attempt to force the model to fit the CS2 estimates by making cumulative changes to the model that are biased towards thickening the sea ice. While these are reasonable for assessing which variables may be incorrectly parameterized, they are not really sensitivity studies; they are rather specific model perturbations intended to produce thicker sea ice. A terminology other than "sensitivity study" should be used.

Suggested change in manuscript: Following your suggestion, we change the header for Section 3.3 "Sensitivity simulations" to "Simulations with perturbed physical parameterizations and sensitivity simulations" and modify text at several places within the manuscript accordingly.

GC4: When comparing the CICE-mw-form-e-sd and subsequent models with CS2 estimates, the snow distributions will be different, assuming W99 is used for CS2 estimates. W99 for the comparison grid cells in April is approximately 30 cm, whereas Figure 7b suggest values of half that thickness; the resulting CS2 sea ice thickness estimates would be substantially thicker. If the substantially thinner snow cover distribution (as per Figure 7b) were to be used for the CS2 estimates, then a direct comparison would be more valid.

Response: CPOM and AWI use W99 snow depth for multi-year ice, but they halve W99 snow depth over firstyear ice (see Section 2). The resulting snow depth is actually closer to our reduced snow depth (CICE-mw-forme-sd) than to our original snow depth (CICE-default). The mean April snow depth (2011-2017) over our selected region for modified W99 as applied for CS-2 estimates (28 cm) is only 1 cm thicker than in CICE-mwform-e-sd (27 cm), but 6 cm thinner than in CICE-default (34 cm). The snow depth has been larger over the last years. We acknowledge the suggestion by the reviewer to use simulated snow depth to derive ice thickness from CS2 freeboard for future studies. Suggested change in manuscript: In the modified manuscript we now present the mean April snow depth over the more relevant period from 2011 to 2017 instead of 2005 to 2014 in Fig.7.

Specific comments:

P2, L3: Please include a short description of CICE, with references.

Suggested change in manuscript: We added a few general sentences in Section 3.1. Be aware that applied physical options are mentioned in Section 3.2 and references given.

P2, L20: Discrimination of sea ice type is also a principle challenge (in order to assign ice densities and snow loads); especially when the effects of snow salinity for first-year sea ice are also considered.

Suggested change in manuscript: Thank you. Added.

P2, L32: Can it be called a "full ice thickness distribution" if values < 0.5 m are not included?

Response: Yes. We do exclude grid points with mean ice thickness below 0.5 m, but if the mean value is above 0.5 m, we include all single data points (full range).

P2, L35: Given that "A realistic ITD is critical for simulating ice growth and ice melt rates correctly", an explanation should be included here on the impact of omitting thin ice points.

Response: Misunderstanding. See above.

P3, L31: In order to compare the uncertainties of CS2 estimates with the uncertainties of CICE model results, please summarize previously reported (if any) uncertainties and biases, for sea ice thickness, of CICE-default.

Response: We agree that it would be nice to have quantitative information about uncertainties in CICE model results. Unfortunately, this is not possible. Physical and empirical parameters are nor very well constrained for many parameterizations in sea ice models and there are uncertainties due to the coupling with the atmospheric and oceanic forcing. Modifying these parameters within their range of uncertainty can result in mean ice thickness changes of more than 1 m. See e.g. a study by Kim, J. G., E. C. Hunke, and W. H. Lipscomb (2006): A sensitivity analysis and parameter tuning scheme for global sea-ice modeling, Ocean Modell.,14, 61–80.

P4, L3: See GC3.

Suggested change in manuscript: Changed.

P4, L25: Multiyear sea ice is mentioned, but please include a statement regarding the accuracy of snow on first-year sea ice.

Suggested change in manuscript: We added a statement saying that climatology is likely to be an overestimate over FYI (Kurtz and Farrell, 2011; Webster et al., 2014).

P4, L26: In "Therefore, we apply a climatology for the CS2 period 2010 to 2017", what is meant by "a climatology"? Is this W99 as per P2, L23?

Response: No, we compare multi-year monthly means of sea ice thickness (Fig. 1 + 5).

Suggested change in manuscript: Text changed to: "Therefore, we will compare multi-year monthly means over the CS2 period 2010 to 2017".

P5, L10: The language used is too strong: e.g. "This indicates that the winter ice growth is underestimated in the model." See GC1. Recommend: "This suggests that the winter ice growth may be underestimated in the model."

Response: Our statement is justified. See our response to GC1.

P5, L10: If the CS2 ice thickness is overestimated when thinner and more-accurately estimated (or underestimated) when thicker (as per Tilling et al., 2018) then this may mitigate the CICE April "underestimation".

Response: See our response to GC1. Overall, Tilling et al (2018) found that there was no significant bias in CS2 estimates of sea ice thickness when compared with independent datasets. As all are derived products it is not possible to say what dataset causes the slight variation in bias with ice thickness, and we cannot assume it's CS2.

P5, L29: This statement is rather broad, given that only air temperature and incoming longwave were assessed, and then only for two specific perturbations. Please reword.

Response: Atmospheric forcing consists of air temperature and humidity, incoming longwave and shortwave radiation, wind vector and precipitation. Shortwave radiation is negligible in winter, the impact of errors in air humidity and wind vector on ice growth is small. Thus, air temperature and incoming longwave radiation are the main atmospheric drivers. Our perturbations (-2 K and -15 Wm-2 everywhere and over the whole model period) are large and represent the full atmospheric impact apart from snowfall.

Suggested change in manuscript: We reword our statement that "the underestimation of winter ice growth cannot be explained by errors in atmospheric forcing data" to "the underestimation of winter ice growth cannot be explained by errors in the surface energy balance associated with atmospheric forcing data".

P6, L11 and Table 1: For CICE-mw-form-e, what is the justification for increasing the longwave emissivity of sea ice from 0.95 to 0.976.

Response: The observed values of longwave emissivity of sea ice ranges from 0.95 to 0.99. The value of 0.976 has recently been applied by Ridley et al. (GMD, 2018) for climate simulations. E.g. the sea ice model LIM uses a value of 0.97.

Technical corrections:

Figure 1 caption: "(mean values for November to April vary between 99.4 and 998. %." Is missing a ")" and 998 is likely 99.8%.

Suggested change in manuscript: Thank you. Modified.

Figure 3: Change North pole latitude to 90, or omit. In the caption, would not "grid cells" be more accurate than "region".

Suggested change in manuscript: Thank you. Changed in all figures.

P3, L26: Define "SIT".

Suggested change in manuscript: We use full word now: sea ice thickness.

Table 1 caption: "...model changes..."

**Suggested change in manuscript: Thank you. Done.**

Table 1: surfaceaography = surface topography?

Suggested change in manuscript: Yes. Changed.

**Anonymous Referee #2**

**General Comments**

In this paper, the authors evaluate the ice thickness from a standalone CICE model forced with NCEP-2 Reanalysis atmospheric forcing against CryoSat-2-derived ice thickness during 2010-2017 to determine the cause of the model's underestimation of the central Arctic winter ice thickness. A multi-year simulation is performed (CICE-free) for the period of 1980-2017 on a 40 km CICE (v5.1.2) grid using a simple mixed layer ocean model where the ocean temperature is restored to climatology on 20-day restor- ing time scale and utilizes monthly mean climatological ocean currents. Model defaults include a prognostic melt pond, elastic anisotropic rheology, ridging and the Delta- Eddington radiation scheme. Comparisons against CryoSat-2 (CS2) sea ice thickness from CPOM, AWI, and NASA for the months of November – April all reveal a significant under-prediction in the winter Central Arctic ice thickness. The CS2 data is binned into five CICE model ice thickness categories ranging from the first category with < 0.6 m of ice to the fifth category with ice > 3.6 m on a rectangular 50 km grid and used to initialize a series of experiments. The paper discusses the treatment of snow for each CS2 dataset in deriving the ice thickness. In their study, the authors chose a region in the Central Arctic (Fig. 3) with a minimum ice thickness of 0.5 m and a region where ice growth is dominant over the impact of sea ice dynamics. A series of 7 studies are performed for the multi-year periods beginning in November of 2010 and continuing for 15 months and repeated through 2016/2017. Experiment are initialized with all three CS2 ice thickness fields and use different CICE options documented in Table 1.

An experiment was performed by decreasing the longwave radiation by 15 w/m2 and reducing the air temperature by 2 K everywhere and for all times. They found some differences in the mean September ice thickness, but little change in winter ice thick- ness; concluding that the under-prediction in winter ice growth can not be explained by errors in forcing data.

Experiments were conducted which studied the impact of increased total ice surface albedo by releasing more melt water into ocean; thus reducing ponded water over the surface of ice. This study (CICE-mw) only revealed a small improvement in winter ice thickness.

Another set of experiments applied the form drag parameterization of Tsamados et al. (2014) to investigate sea ice advection and turbulent heat fluxes (CICE-mw-form). They found a small ice thickness increase attributed to a reduce ice drift speed resulting in a weaker ocean-ice heat flux. Another set of experiments increased the emissivity of snow from 0.95 to 0.976 (CICE-mw-form-e) impacting the longwave radiation budget. They found that the summer melt was reduced by only a few cm, but no noticeable change was found in winter.

The most significant improvement was found with the implementation of a snow drift scheme based on Lecomte et al. (2014) which reduced the snow depth by 20-40% (CICE-mw-form-e-sd) which lowered the winter ice thickness errors from 0.8 to 0.25 m. They found that the reduction of snow led to larger winter ice growth as well as an increase in the summer melt due to the earlier disappearance of snow. A final set of studies (CICE-mw-form-e-sd-bubbly) increased the conductivity for colder temperatures (Pringle et al., 2007) and further reduced the errors to less than 0.1m. This configuration is referred to as "CICE-best".

Using the CICE-best configuration, the authors studied the length of the melt season and found a 7-day reduction from 107 to 100 days versus the "observed" value of 94 days. They performed studies using "cold

forcing from the 1980's" which led to an increase in the September ice thickness of 0.8 m, but only 0.11 m in April.

The authors conclude that warm winters are not responsible for the observed sea ice thinning in the Central Arctic over the past decade; rather it depends strongly on atmospheric conditions from May to June in particular, with the start of the melt season and formation of melt ponds, thus preconditioning the strength of the positive albedo feed- back. They found that the recommended improvements to the CICE model outweigh the impact of initialization with CS2 ice thickness fields.

This is a very well written and thorough paper with carefully designed model experiments. I highly recommend it for publication.

Response: We would like to thank the reviewer for the comprehensive summary and the positive feedback.

**Specific Comments**

Throughout the paper your refer to 1-year experiments with the CS2 data, but your time series plots look like the experiments run from (for example) Nov 2010 – January 2012, so a 15-month experiment?

Response: Ini-runs were performed over 1-year. In Fig. 1b and 5 results from November to October are shown. November to February results have just been repeated to visualize annual cycle.

Suggested change in manuscript: We added note in figure caption.

The CICE tripole grid has a resolution of  $\sim$ 40 km. Why wasn't the CS2 data interpolated onto that same 40km grid (instead of 50 km)?

Response: CS2 ice thickness distribution has first been interpolated on a regular 50 km grid to facilitate usage of other groups and afterwards re-gridded on ORCA1deg grid. The Impact of grid transformation on results is marginal.

Please review the use of colors chosen for Figures 1, 5, 8, 10, and 11. Some colors are a bit difficult to differentiate. Perhaps including dashed lines and reducing the number of colors could help.

Suggested change in manuscript: Line colour and style have been adjusted for all these Figures.

**Technical Corrections**

Page 2 line 2: replace "Blockley et al" to "Blockley and Peterson"

Suggested change in manuscript: Thank you. Done.

Page 2 line 28: Spell out ORCA

Response: ORCA refers to the grid defined in Madec, G. and M. Imbard (1996), "A global ocean mesh to overcome the north pole singularity", Climate Dynamics, Vol. 12, p381-388, but no information about meaning available in literature.

Page 2 lines 33,34: replace "Libscomp" to "Lipscomb"

Suggested change in manuscript: Thank you. Corrected.

Page 5 line 20: add "by 2 K during the whole simulation. . ."

Suggested change in manuscript: Added.

Page 6 line 2: replace "satellite products" to "satellite products (not shown)"

Suggested change in manuscript: Added.

Page 8 line 21: should be "Maaß"

Suggested change in manuscript: Thank you. Corrected.

Page 13 Table 1 caption: should be "Note that all model changes. . ."

Suggested change in manuscript: Thank you. Corrected.

Page 15 Figure 1 caption: replace "998." With "99.8%"

Suggested change in manuscript: Thank you. Corrected.

Page 22: Why are AWI/NASA both showing up as solid grey lines? Can one be shown as dashed?

Suggested change in manuscript: Yes. Adjusted.

Page 24, 25: same question as above

Suggested change in manuscript: Yes. Adjusted.

[revised manuscript text omitted]

| CICE-mwThe maximum melt water added to melt ponds fracmax is reduced from 10% to
YYYYObs: The actual fraction of mut water fractice actualted as: fraction is the fraction in the fraction in 1%. This reduction accounts for the uncertainty in the fraction of mut water that collects in ponds versus being immediately released to the occan. The impact of this change on the simulated melt pond fraction in 10½ 2012 is demonstrated to Fig. 1. The melt pond fractions (Reesel et al., 2012).YCICE-mw-formInstead of a constant drag coefficient for the momentum fluxes between atmosphere and ice (CDa = 1.3 x 10-3) and between ice and occan (CDb = 5.36 x 10-3), the form drag parametrization of 1 samadas et al. (2014) is applied accounting for the impact of pressure ridges, keels, ice floe and melt pond edges. Here, we modify the background drag coefficient for the atmosphere (csa = 0.01) instead of 0.02) and the occan (csw = 0.0006) instead of 0.02) and the parameters determining the impact of ridges and keels (cra = 0.1) instead 0.02 and erw = 0.5YCICE-mw-form-eThe longwave emissivity of sea ice is increase ice drift over level ice and decrease ice drift over level dice costiling in a more realistice dor float time in comparison to Pathfinder (not shown).YYCICE-mw-form-e-sdDepending on wind speed, snow density and surface longsorphy, snow can be eracted from the saic variate, and the realistical or los in iterastical or los in iterastical or los in iterastical or los in iterastical or los in the east ice straft action in one water straft action in the cast is applied account for these processes. Here, we eparameters de since water straft action and be realistical or los in iterastical or los in iterastical or los in a los of 0.25 for ice attagory 1.4 m < A < 2.4 m; W apply a Autar = 3.5 m s^2, current swind speed is the straft a                                                                                                                                                                                                                                                                                                                                                           | Run Name                                | Description                                                                                                                                                                                                                                                                                                                                                                                                                                                                                                                                                                                                                                                                                                                                                                                                                                                                                                                                                                                                                                                                                                                                                                                                                                                                                                                                                                                                                                                                                                                                                                                                                                                                                                                                                                                                                                                                                                                                                                                                                                                                                                                                                                                                                                                                            | free | ini |
|------------------------------------------------------------------------------------------------------------------------------------------------------------------------------------------------------------------------------------------------------------------------------------------------------------------------------------------------------------------------------------------------------------------------------------------------------------------------------------------------------------------------------------------------------------------------------------------------------------------------------------------------------------------------------------------------------------------------------------------------------------------------------------------------------------------------------------------------------------------------------------------------------------------------------------------------------------------------------------------------------------------------------------------------------------------------------------------------------------------------------------------------------------------------------------------------------------------------------------------------------------------------------------------------------------------------------------------------------------------------------------------------------------------------------------------------------------------------------------------------------------------------------------------------------------------------------------------------------------------------------------------------------------------------------------------------------------------------------------------------------------------------------------------------------------------------------------------------------------------------------------------------------------------------------------------------------------------------------------------------------------------------------------------------------------------------------------------------------------------------------------------------------------------------------------------------------------------------------------------------------------------------------------|-----------------------------------------|----------------------------------------------------------------------------------------------------------------------------------------------------------------------------------------------------------------------------------------------------------------------------------------------------------------------------------------------------------------------------------------------------------------------------------------------------------------------------------------------------------------------------------------------------------------------------------------------------------------------------------------------------------------------------------------------------------------------------------------------------------------------------------------------------------------------------------------------------------------------------------------------------------------------------------------------------------------------------------------------------------------------------------------------------------------------------------------------------------------------------------------------------------------------------------------------------------------------------------------------------------------------------------------------------------------------------------------------------------------------------------------------------------------------------------------------------------------------------------------------------------------------------------------------------------------------------------------------------------------------------------------------------------------------------------------------------------------------------------------------------------------------------------------------------------------------------------------------------------------------------------------------------------------------------------------------------------------------------------------------------------------------------------------------------------------------------------------------------------------------------------------------------------------------------------------------------------------------------------------------------------------------------------------|------|-----|
| CICE-mw-form Instead of a constant drag coefficient for the momentum fluxes between at mosphere and ice (CDa = 1.3 x 10-3) and between ice and occan (CDe 3.36 x 10-3), the form drag parametrization of Tsamados et al. (2014) is applied accounting for the impact of pressure ridges, keels, ice flow and melt pond edges. Here, we modify the background drag coefficient for the atmosphere (csa = 0.01 instead of 0.03) and the occan (csw = 0.0005 instead of 0.02) and the parameters determining the impact of ridges and keels (cra = 0.1) instead of 0.2.1 mese modifications increase ice drift over level ice and decrease ice drift over ridged ice resulting in a more realistic ice drift pattern in comparison to Pathfinder (not shown). The longwave emissivity of sea ice is increased from 0.95 to 0.976. Y Y CICE-mw-form-e-B Depending on wind speed, snow density and surface longwarpathy, snow can be eroded from the sea ice surface, drift through air and be redistributed to lost in leads. The default CICE simulation does not clocount for these processes. Here, we parameterize the snow erosion rate following Lecomte et al. (2014): $\frac{\partial h_s}{\partial t} = -\frac{\gamma}{\sigma_{TD}} (V - V^*) \frac{\rho_{SMAX} - \rho_S}{\rho_S}$ with snow depth hr, mass flux tuning coefficient $\gamma = 10^3$ kg m², current wind speed V 4 = 3.5 m s 4 , current snow density $\rho$ and maximum snow density $\rho_{ALM,A}$ and standard deviation of ice thickness distribution, we apply $\rho_{ALM,T} = 330$ kg m² (the constant snow density row and $\rho$ mod J m for category 3 1.4 m < h < 2.4 m). We assume that the whole amount of snow blown into the air will be released into the occan. Estimating the error of this assumption, we calculate the net source - ading the blow wind speed stays above the threshold value. Assuming an average friction velocity of 1. m s -1 and rate in adi distance O 200 m, one cycle takes approximately 30 min. For every cycle, the lead fraction defines the fraction of snow blown into the air will be released into the occan. Stimuting the error of this assumption, we calculate                                                         | CICE-mw                                 | The maximum m elt water added to melt ponds rfracmax is reduced from 100% to 50%. The actual fraction of melt water rfrac is calculated as: rfrac = rfracmin + (rfracmax-rfracmin) * aice with rfracmin = 15%. This reduction accounts for the uncertainty in the fraction of melt water that collects in ponds versus being immediately released to the ocean. The impact of this change on the simulated melt pond fraction in July 2012 is demonstrated in Fig. 1. The melt pond fraction in the central Arctic is generally reduced by 5-10% ranging from 25 to 40% in the default simulation and from 20 to 35% in CICE mw. The new melt pond fractions (Roesel et al., 2012).                                                                                                                                                                                                                                                                                                                                                                                                                                                                                                                                                                                                                                                                                                                                                                                                                                                                                                                                                                                                                                                                                                                                                                                                                                                                                                                                                                                                                                                                                                                                                                                             | Y    | Y   |
| CICE-mw-form-eThe longwave emissivity of sea ice is increased from 0.95 to 0.976.YYCICE-mw-form-e-sdDepending on wind speed, snow density and surface topaography, snow can be
eroded from the sea ice surface, drift through air and be redistributed or lost in
leads. The default CICE simulation does not account for these processes. Here,
we parameterize the snow crosion rate following Lecomte et al. (2014):YY $\frac{\partial h_s}{\partial t} = -\frac{\gamma}{\sigma_{TD}} (V - V^*) \frac{\rho_{s,MAX} - \rho_s}{\rho_s}$
with snow depth h, mass flux tuning coefficient $\gamma = 10^5$ kg m², current wind
speed V, threshold wind speed $V^* = 3.5$ m s¹, current snow density $\rho_i$ and
maximum snow density $\rho_{A,MAX}$ , and standard deviation of ice thickness
distribution $\sigma_{TD}$ . Lacking information about the snow density in tribution, we
apply $\rho_{A,MAX} = 330$ kg m³ (the constant snow density in CICE) and assume $\rho_i =$
240 kg m². Regarding the ITD, we apply $\sigma$ values of 0.25 m for ice category 1
(ice thickness h < 0.6 m) 0.5 m for category 2 (0.6 m < h < 1.4 m) and 1 m for
category 3 1.4 m < h < 2.4 m). We assume that the whole amount of snow blown
into the air will be cleased into the ocean. Estimating the error of this
assumption, we calculate the net snow re-deposition rate. Snow which is blown
into air, will be deposited at the surface and might be blown into the air again if
the wind speed stays above the threshold value. Assuming an average friction
velocity 0 0.1 m s² in and a total distance of 200 m, one cycle takes approximately
30 min. For every cycle, the lead fraction defines the fraction of snow volume
ereleased into the ocean. Analyzing NCEP-2 wind fields, the average period the
wind speed stays above the threshold value close to coast of North Greenland.
Thus, for most parts of the Arctic more than 90 % of the total snow blown into the
air would be lost in leads. A                                                                                                                                                                                                          | CICE-mw-form                            | Instead of a constant drag coefficient for the momentum fluxes between
atmosphere and ice (CDa = $1.3 \times 10-3$ ) and between ice and ocean (CDo = $5.36 \times 10-3$ ), the form drag parametrization of Tsamados et al. (2014) is applied
accounting for the impact of pressure ridges, keels, ice floe and melt pond edges.
Here, we modify the background drag coefficient for the atmosphere (csa = $0.01$
instead of $0.005$ ) and the ocean (csw = $0.0005$ instead of $0.02$ ) and the parameters
determining the impact of ridges and keels (cra = $0.1$ instead of $0.2$ and crw = $0.5$
instead of $0.2$ ). These modifications increase ice drift over level ice and decrease
ice drift over ridged ice resulting in a more realistic ice drift pattern in comparison
to Pathfinder (not shown).                                                                                                                                                                                                                                                                                                                                                                                                                                                                                                                                                                                                                                                                                                                                                                                                                                                                                                                                                                                                                                                                                                                                                                                                                                                                                                                                                                                                                                         | Y    | Y   |
| CICE-mw-form-e-sd Depending on wind speed, snow density and surface topography , snow can be coded from the sea ice surface, drift through air and be redistributed or lost in leads. The default CICE simulation does not account for these processes. Here, we parameterize the snow erosion rate following Lecomte et al. (2014):
$\frac{\partial h_s}{\partial t} = -\frac{\gamma}{\sigma_{ITD}} (V - V^*) \frac{\rho_{s,MAX} - \rho_s}{\rho_s}$ with snow depth $h_r$ , mass flux tuning coefficient $\gamma = 10^2$ kg m 2 , current wind speed $V$ , threshold wind speed $V^* = 3.5$ m s -1 , current snow density $\rho_s$ and maximum snow density $\gamma_{s,MAX}$ , and standard deviation of ice thickness distribution $\sigma_{TD}$ . Lacking information about the snow density distribution, we apply $\rho_{AMAX} = 330$ kg m 3 (the constant snow density in CICE) and assume $\rho_{T} = 240$ kg m 3 . Regarding the ITD, we apply $\sigma$ values of 0.25 m for ice category 1 (ice thickness $h < 0.6$ m), 0.5 m for category 2 (0.6 m $< h < 1.4$ m) and 1 m for category 3 1.4 m $< h < 2.4$ m). We assume that the whole amount of snow blown into the air will be released into the ocean. Estimating the error of this assumption, we calculate the net snow re-deposition rate. Snow which is blown into the air will be exposited at the surface and might be blown into the air again if the wind speed stays above the threshold value. Assuming an average friction velocity of 0.1 m s -1 and a total distance of 200 m, one cycle takes approximately 30 min. For every cycle, the lead fraction defines the fraction of snow volume released into the ocean. Analyzing NCEP-2 wind fields, the average period the wind speed stays above the threshold value of 3.5 m s 2 ranges from 50 to 120 h over the Arctic sea ice with the lowest values close to coast of North Greenland. Thus, for most parts of the Arctic more than 90 % of the total snow blown into the air would be lost in leads. An error of less than 10 % justifies our simplification. The impact of our parameterization of more through word wetph words by each winto the air would be lost | CICE-mw-form-e                          | The longwave emissivity of sea ice is increased from 0.95 to 0.976.                                                                                                                                                                                                                                                                                                                                                                                                                                                                                                                                                                                                                                                                                                                                                                                                                                                                                                                                                                                                                                                                                                                                                                                                                                                                                                                                                                                                                                                                                                                                                                                                                                                                                                                                                                                                                                                                                                                                                                                                                                                                                                                                                                                                                    | Y    | Y   |
| CICE-mw-form-e-sd-bubbly We apply the bubbly conductivity formulation from Pringle et al. (2007) which Y Y (CICE-best) results in larger thermal conductivity values for colder ice temperatures                                                                                                                                                                                                                                                                                                                                                                                                                                                                                                                                                                                                                                                                                                                                                                                                                                                                                                                                                                                                                                                                                                                                                                                                                                                                                                                                                                                                                                                                                                                                                                                                                                                                                                                                                                                                                                                                                                                                                                                                                                                                            | CICE-IIIW-IOIIII-e-su                   | Depending on which speed, show density and surface top adgraphy, show can be eroded from the sea ice surface, d rift through air and be redistributed or lost in leads. The default CICE simulation does not account for these processes. Here, we parameterize the snow erosion rate following Lecomte et al. (2014):
$\frac{\partial h_s}{\partial t} = -\frac{\gamma}{\sigma_{ITD}} (V - V^*) \frac{\rho_{s,MAX} - \rho_s}{\rho_s}$ with snow depth h s, mass flux tuning coefficient $\gamma = 10^{-5}$ kg m -2 , current wind speed V , threshold wind speed $V^* = 3.5$ m s -1 , current snow density $\rho_s$ and maximum snow density $\rho_{s,MAX}$ , and standard deviation of ice thickness distribution $\sigma_{ITD}$ . Lacking information about the snow density distribution, we apply $\rho_{s,MAX} = 330$ kg m 3 (the constant snow density in CICE) and assume $\rho_s = 240$ kg m 3 . Regarding the ITD, we apply $\sigma$ values of 0.25 m for ice category 1 (ice thickness $h < 0.6$ m), 0.5 m for category 2 (0.6 m < $h < 1.4$ m) and 1 m for category 3 1.4 m < $h < 2.4$ m). We assume that the whole amount of snow blown into the air will be released into the occan. Estimating the error of this assumption, we calculate the net snow re-deposition rate. Snow which is blown into air, will be deposited at the surface and might be blown into the air again if the wind speed stays above the threshold value. Assuming an average friction velocity of 0.1 m s -1 and a total distance of 200 m, one cycle takes approximately 30 min. For every cycle, the lead fraction defines the fraction of snow volume released into the lowest values close to coast of North Greenland. Thus, for most parts of the Arctic more than 90 % of the total snow blown into the air would be lost in leads. An error of less than 10 % justifies our simplification. The impact of our parameterization on the simulated snow depth can be seen in Fig. 2. Accounting for the loss of drifting snow reduces the snow depth between 20 and 40 %. The larger differences occur over regions with strongest winds and the smallest differences north of Greenland and Cana | I    | I   |
|                                                                                                                                                                                                                                                                                                                                                                                                                                                                                                                                                                                                                                                                                                                                                                                                                                                                                                                                                                                                                                                                                                                                                                                                                                                                                                                                                                                                                                                                                                                                                                                                                                                                                                                                                                                                                                                                                                                                                                                                                                                                                                                                                                                                                                                                                    | CICE-mw-form-e-sd-bubbly
(CICE-best) | We apply the bubbly conductivity formulation from Pringle et al. (2007) which results in larger thermal conductivity values for colder ice temperatures.                                                                                                                                                                                                                                                                                                                                                                                                                                                                                                                                                                                                                                                                                                                                                                                                                                                                                                                                                                                                                                                                                                                                                                                                                                                                                                                                                                                                                                                                                                                                                                                                                                                                                                                                                                                                                                                                                                                                                                                                                                                                                                                        | Y    | Y   |

712. **Table 2**. Sensitivity simulations exploring the impact of uncertainty in atmospheric forcing data. "Free" indicates multi-year simulations from 1980 to 2017; "ini" indicates seven 1-year-long simulations starting in mid-November with CS2 sea ice thickness (2010/2011 to 2016/17).

| Run Name     | Description                                                                     | free | ini |
|--------------|---------------------------------------------------------------------------------|------|-----|
| CICE-Ldown15 | As CICE-default, but forcing field incoming longwave radiation has been         | Ν    | Y   |
|              | decreased by 15 % everywhere and for all times.                                 |      |     |
| CICE-Tair2   | As CICE-default, but forcing field 2m-air temperature has been decreased by 2 K | Ν    | Y   |
|              | everywhere and for all times.                                                   |      |     |

5

**712. <del>712.</del>713.**

[revised manuscript text omitted]